# Harnessing Uncertainty: Entropy-Modulated Policy Gradients for Long-Horizon LLM Agents

## Abstract

In long-horizon tasks, recent agents based on Large Language Models (LLMs) face a significant challenge that sparse, outcome-based rewards make it difficult to assign credit to intermediate steps. Previous methods mainly focus on creating dense reward signals to guide learning, either through traditional reinforcement learning techniques like reward shaping and intrinsic motivation or by using Process Reward Models for step-by-step feedback. In this paper, we identify a fundamental problem in the learning dynamics of LLMs: the magnitude of policy gradients is inherently coupled with the entropy, which leads to inefficient small updates for confident correct actions and potentially destabilizes large updates for uncertain ones. To resolve this, we propose Entropy-Modulated Policy Gradients (EMPG), a framework that re-calibrates the learning signal based on step-wise uncertainty and the final task outcome. EMPG amplifies updates for confident correct actions, penalizes confident errors, and attenuates updates from uncertain steps to stabilize exploration. We further introduce a bonus term for future clarity that encourages agents to find more predictable solution paths. Through comprehensive experiments on three challenging agent tasks, WebShop, ALFWorld, and Deep Search, we demonstrate that EMPG achieves substantial performance gains and significantly outperforms strong policy gradient baselines.

## 1 Introduction

The advent of Large Language Models (LLMs) has catalyzed the development of autonomous agents that are capable of tackling complex, multi-step tasks (Wei et al., 2022; Yao et al., 2023). However, a fundamental challenge persists in training these agents for long-horizon tasks: the sparsity of outcome-based rewards. In many realistic scenarios, such as web navigation (Yao et al., 2022), software engineering Zhang et al. (2024), and deep search (Alzubi et al., 2025), feedback is only available at the end of the complete generation. This makes it difficult to assign appropriate credit for standard reinforcement learning (RL) algorithms to discern the crucial intermediate steps.

To address sparse rewards, prior work has explored either densifying reward signals via techniques like reward shaping and intrinsic motivation, or providing explicit step-wise supervision with Process Reward Models (PRMs) (Lightman et al., 2023). Both approaches face significant hurdles. Reward densification methods often fail to scale to the vast state-action spaces of LLM agents, while PRMs are prohibitively expensive to annotate, struggle with generalization, and are impractical for complex interactive tasks where defining a single "correct" intermediate step is often impossible.

Policy entropy has also been repurposed as a learning signal. Some methods use entropy minimization as an unsupervised objective to increase model certainty (Gao et al., 2025; Agarwal et al., 2025), but risk inducing "hallucinated confidence" where the model becomes confidently incorrect Zhang et al. (2025d). More recent work uses entropy to modulate the learning signal in single-turn, reasoning tasks (Chen et al., 2025; Cheng et al., 2025). However, it remains underexplored how to leverage an agent's intrinsic uncertainty for credit assignment in long-horizon, multi-step decision-making.

Our work begins by analyzing the fundamental dynamics of the policy gradient itself. We formally show that for a standard softmax policy, the expected norm of the score function is a monotonic function of the policy's entropy (Proposition 1). In simple terms, high-entropy (uncertain) actions

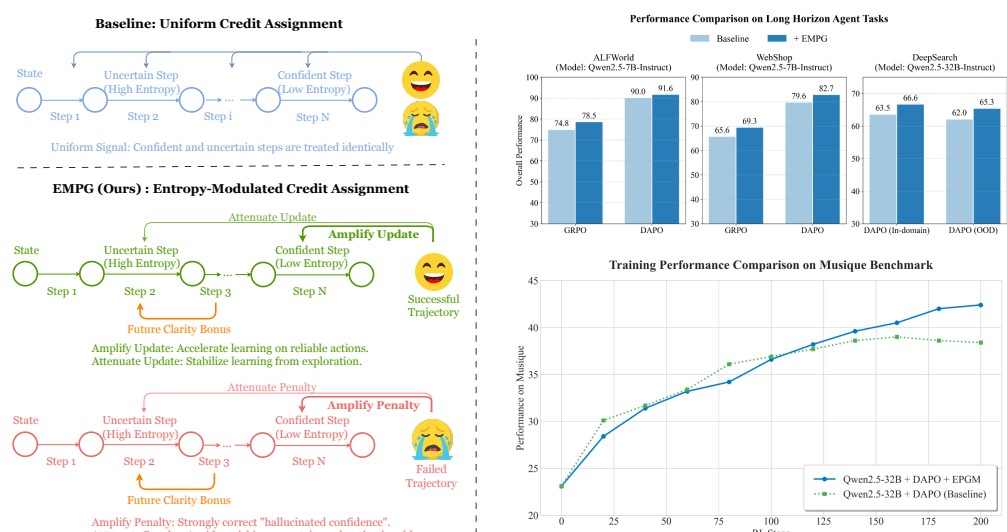

Figure 1: Overview of the EMPG mechanism and its algorithm performance. **Left:** Conceptual diagram contrasting the uniform credit assignment of baseline methods with EMPG's confidence-modulated signal. **Right:** Final performance comparison on key long-horizon benchmarks showing EMPG's superiority, along with the training dynamics on Musique that highlight its ability to achieve sustained improvement and avoid the baseline's performance plateau.

naturally produce large gradients, while low-entropy (confident) actions produce small ones. This inherent behavior presents a dual challenge for learning: 1) confident and correct steps, which should be strongly reinforced, receive small updates, limiting learning speed, and 2) uncertain exploratory steps can introduce large, noisy gradients that destabilize training. This reveals a critical need to explicitly re-calibrate the learning signal based on an action's uncertainty.

To address this, we propose Entropy-Modulated Policy Gradients (EMPG), a framework that reshapes the learning landscape by directly adapting to this dynamic, as illustrated in Figure 1. Instead of naively rewarding low entropy, EMPG introduces *Self-Calibrating Gradient Scaling* mechanism, which dynamically modulates the policy gradient based on step-wise uncertainty: *1) for confident and correct actions*, it amplifies the updates, while *2) for uncertain steps*, it attenuates updates to ensure stable exploration. Furthermore, to encourage agents to find predictable solution paths, EMPG introduces *"future clarity"*, an additional bonus term in the advantage function that provides an intrinsic signal for actions that lead to less uncertain subsequent states. This guides agents to perform purposeful exploration, steering them away from chaotic or unpromising high-entropy trajectories toward states with greater clarity about the next steps. This dual approach enables EMPG to forge a dense, informative, and well-calibrated learning signal from sparse external feedback. To validate our framework, we conduct experiments on challenging long-horizon agent benchmarks such as WebShop Yao et al. (2022), ALFWorld Shridhar et al. (2021), and Deep Search Alzubi et al. (2025), demonstrating the effectiveness and scalability of our approach across models of various sizes.

Our key contributions are as follows:

- We first identify and formalize a fundamental challenge in policy gradient methods: the inherent coupling of gradient magnitude and policy entropy. This dynamic leads to inefficient learning for confident actions and instability from uncertain ones, motivating the need for explicit signal re-calibration.
- We introduce Entropy-Modulated Policy Gradients, a framework designed to solve this problem. EMPG combines *Self-Calibrating Gradient Scaling* to correct the flawed gradient dynamics with a *Future Clarity Bonus* to promote exploration towards more predictable states.
- Extensive experiments on demanding agent tasks (WebShop, ALFWorld, Deep Search) show that EMPG substantially outperforms strong baselines like GRPO and DAPO.

## 2 RELATED WORK

### 2.1 LLM-BASED AUTONOMOUS AGENTS

The advent of LLMs has catalyzed the development of sophisticated autonomous agents capable of performing complex, multi-step tasks that were previously unattainable. Specialized agents have been designed for diverse applications, including software development (e.g., coding agents (Zhang et al., 2024)), information retrieval (search agents (He et al., 2025; Li et al., 2025)), and complex web interactions (browser-use agents (Yao et al., 2022; Deng et al., 2023; Yan et al., 2023)). For training these agentic models, reinforcement learning has proven to be a powerful and essential paradigm. Recent research on RL-based agents, such as Search-R1 (Jin et al., 2025), SWE-RL (Wei et al., 2025a), and WebAgent-R1 (Wei et al., 2025b), has demonstrated that RL can effectively enhance agent performance and enable learning in highly interactive and dynamic environments. Despite these successes, a fundamental problem remains to be fully addressed: the difficulty of credit assignment in long-horizon tasks. The multi-step nature of these problems, where a reward signal is often only available upon completion, hinders the efficiency and stability of the training process.

### 2.2 REINFORCEMENT LEARNING FROM INTERNAL FEEDBACK

To overcome the challenges of sparse external rewards, recent studies have explored using internal feedback, generated by the model itself, to create denser training signals. This approach often leverages unsupervised signals derived from model uncertainty (Zhang et al., 2025b; Agarwal et al., 2025; Zhao et al., 2025) or self-consistency (Zuo et al., 2025; Zhang et al., 2025a), frequently quantified by policy entropy. However, the role of entropy has been interpreted in conflicting ways. Some studies argue that correct responses typically exhibit lower entropy, thus proposing unsupervised entropy minimization as a method to improve performance (Gao et al., 2025); for example, Agarwal et al. (2025) focuses on minimizing the entropy of the entire generated trajectory to enhance the confidence and quality of the final output, typically in single-turn reasoning tasks. Conversely, other works suggest that high entropy encourages exploratory reasoning. For instance, SEED-GRPO (Chen et al., 2025) uses semantic entropy to modulate policy updates for diversity, while others explicitly incorporate policy entropy into the advantage term to promote exploration (Cheng et al., 2025; Vanlioglu, 2025). Recently, EDGE-GRPO (Zhang et al., 2025c) proposes entropy modulation in single-turn mathematical reasoning. Similar to our method, they modulate policy gradients by amplifying updates for confident correct responses and attenuating updates for incorrect or uncertain ones. However, EMPG fundamentally differs from EDGE-GRPO in both motivation and scope: First, while EDGE-GRPO focuses on correcting confidence misalignment within a single-turn mathematical reasoning, EMPG is specifically designed for the multi-step credit assignment problem in long-horizon tasks. Second, towards the challenges in multi-turn long-horizon tasks, EMPG dynamically assigns credit across the entire trajectory to amplify the crucial steps.

## 3 PRELIMINARIES

### 3.1 BACKGROUND: POLICY OPTIMIZATION FOR LONG-HORIZON AGENT TASKS

We formalize the long-horizon agent task as a reinforcement learning problem where an LLM-based policy, $\pi_\theta$, is optimized to maximize the expected total return, $R(\tau)$. A foundational approach in this domain is Proximal Policy Optimization (PPO), which ensures training stability by using a learned value model to estimate step-wise advantages (Schulman et al., 2017). However, this approach introduces substantial memory and computational overhead. Furthermore, its effectiveness hinges on value estimates that are difficult to learn accurately, especially in sparse-reward, long-horizon tasks.

Due to these challenges, value-free policy gradient methods have become a popular and effective paradigm, as they avoid the overhead and instability of a learned value function (Shao et al., 2024; Yu et al., 2025). Methods like Group Relative Policy Optimization (GRPO) and Decoupled Clip and Dynamic Sampling Policy Optimization (DAPO) provide robust credit assignment by comparing multiple trajectory rollouts. While effective at avoiding value model pitfalls, these strategies still rely on coarse, trajectory-level credit assignment. This fails to pinpoint critical actions and ignores the rich, intrinsic signal of the model's own step-wise uncertainty—the very signal our work leverages. Details are provided in Appendix B.

## 3.2 Theoretical Motivation: A Two-Part Re-Calibration of Policy Gradients

Our approach is motivated by a fundamental analysis of the relationship between a policy's gradient and its predictive uncertainty. Standard policy gradients, while effective, possess an inherent dynamic that can hinder stable and efficient learning. Specifically, the magnitude of the gradient is inherently coupled with the policy's entropy, often leading to inefficiently small updates for confident actions and potentially destabilizing large updates for uncertain ones. This dynamic, which we aim to re-calibrate, is formally characterized by the following proposition.

**Proposition 1.** *For a policy $\pi_\theta$ parameterized by a softmax over logits $z_\theta(s)$, the expected squared L2-norm of the score function $\nabla_{z_\theta} \log \pi_\theta(a|s)$ with respect to the logits is a direct function of the policy's Rényi-2 entropy Rényi (1961), $H_2(\pi)$:*

$$\mathbb{E}_{a \sim \pi_\theta(\cdot|s)} \left[ ||\nabla_{z_\theta(s)} \log \pi_\theta(a|s)||^2 \right] = 1 - \exp(-H_2(\pi_\theta(\cdot|s))) \tag{1}$$

*A detailed proof is provided in Appendix C.*

Equation 1, which builds upon established relationships between different measures of policy entropy (e.g., in Li (2025)), proves that the expected gradient norm is monotonically coupled with policy entropy. This presents a dual challenge: *1)* a confident and correct step should be reinforced strongly, but its naturally small gradient limits its impact; and *2)* the large gradients from highly uncertain exploratory steps can introduce noise and destabilize training. Our first component, *Self-Calibrating Gradient Scaling*, directly addresses this by re-calibrating the *magnitude* of the update based on current-step uncertainty.

However, re-calibrating the update magnitude is only half the solution. A truly effective learning signal must also guide the agent in a useful *direction*. This motivates our second component, the *Future Clarity Bonus*, which can be conceptually justified through the lens of information theory. The Future Clarity Bonus is formulated as a *step-wise intrinsic reward* that encourages the agent to select actions $a_t$ that lead to *low-entropy (high clarity)* subsequent states $s_{t+1}$. By rewarding the immediate clarity gained, the bonus encourages actions that yield high *Information Gain* about the optimal future path. Crucially, this is a local, step-wise objective aimed at minimizing the policy's entropy *at the next state*, rather than minimizing the entropy of the full trajectory:

$$\min_{a_t} H\big(\pi_\theta(\cdot|s_{t+1})\big). \tag{2}$$

This objective, which aligns with established principles like the Empowerment framework Klyubin et al. (2005), imbues the agent with a generalizable meta-skill: to actively seek clarity in the face of ambiguity, effectively turning the complex problem at $s_t$ into a "more solvable" or "less ambiguous" sub-problem at $s_{t+1}$.

In summary, EMPG provides a complete, two-part re-calibration of the learning signal. The gradient scaling module ensures each update has an appropriate *magnitude*, while the future clarity bonus provides a principled intrinsic motivation that shapes the policy's *direction* towards robust and predictable solution paths.

## 4 Entropy-Modulated Policy Gradients

Building on the theoretical motivation established in our preliminaries, we introduce Entropy-Modulated Policy Gradients (EMPG), a framework designed to re-calibrate the learning dynamics of policy gradients for long-horizon agent tasks. As shown in Section 3.2, standard policy gradients are inherently biased towards applying smaller updates to confident (low-entropy) steps and larger updates to uncertain (high-entropy) ones. EMPG is engineered to counteract this behavior, enabling more efficient and stable learning from sparse, outcome-based rewards.

### 4.1 Quantifying Step-Level Uncertainty

The core of our method is to quantify the agent's confidence at each decision-making step. While various uncertainty measures exist, we opt for a practical and computationally efficient proxy: the average token-level entropy over a single "reason-then-act" step. For a step $step_t$ composed of tokens

$\{w_1, ..., w_m\}$, the step-level entropy $H_t$ is:

$$H_t = -\frac{1}{m} \sum_{j=1}^{m} \sum_{v \in V} p(v|w_{<j}) \log p(v|w_{<j}) \tag{3}$$

where $p(v|w_{<j})$ is the probability of token $v$ from the vocabulary $V$, as provided by the LLM's policy $\pi_\theta$. A lower $H_t$ indicates higher confidence in the generated step, corresponding to a lower-entropy state in the sense of Proposition 1. This Shannon entropy formulation is utilized as a robust and efficient proxy because, for any distribution, it is monotonically related to the Rényi-2 entropy $(H(\pi) \geq H_2(\pi))$, thus tracking the same core uncertainty principle.

While we use policy entropy for its computational efficiency, future work could explore alternative uncertainty estimators, such as those derived from Monte Carlo dropout or the variance in logits from an ensemble of model heads. However, we believe entropy provides the most direct link to the gradient dynamics analyzed in Proposition 1, making it the most theoretically grounded choice for our framework.

## 4.2 THE MODULATED ADVANTAGE FOR GRADIENT RE-CALIBRATING

In the sparse reward setting, a standard RL advantage function provides a uniform learning signal for all steps within a single trajectory. While simple, this approach overlooks the varying contributions of different steps and their impact on learning stability. To address this, we introduce a novel, modulated advantage estimate, $A_{\text{mod}}$, for each step $t$ in a trajectory $\tau_i$:

$$A_{\text{mod}}(i, t) = \underbrace{A^{(i)} \cdot g(H_t^{(i)})}_{\text{self-calibrating gradient scaling}} + \underbrace{\zeta \cdot f(H_{t+1}^{(i)})}_{\text{future clarity bonus}} \tag{4}$$

This formulation fundamentally re-calibrates the learning signal through two complementary forms of **advantage shaping**. The first term utilizes a step-level entropy-based function $g(H_t^{(i)})$ to dynamically reweight the trajectory's shared advantage $A^{(i)}$, thereby achieving a more granular and confidence-aware gradient update. The second term, a **future clarity bonus**, is an additive shaping signal that encourages the agent to select actions that lead to a more predictable and less ambiguous future state. Together, these two mechanisms transform a coarse, trajectory-level signal into a rich and precise learning signal for each step, which we analyze further in the following sections.

**Self-Calibrating Gradient Scaling** $g(H)$. To counteract the natural gradient dynamics, the scaling function $g(H)$ is designed to be self-calibrating and adaptive. It achieves this by enforcing the constraint that the mean of $g(H_t^{(i)})$ over any given mini-batch is normalized to one. Mathematically, for a mini-batch of size $N_B$, this constraint is given by:

$$\frac{1}{\sum_{i=1}^{N_B} T_i} \sum_{i=1}^{N_B} \sum_{t=1}^{T_i} g(H_t^{(i)}) = 1 \tag{5}$$

This principled design ensures the modulation redistributes the learning signal rather than simply inflating or deflating it, offering stability, adaptivity, and a reduction in hyperparameters. We implement this by normalizing a base exponential function by its mean over the mini-batch:

$$g(H_t^{(i)}) = \frac{\exp(-k \cdot H_{\text{norm},t}^{(i)})}{\frac{1}{\sum_{j=1}^{N_B} T_j} \sum_{j=1}^{N_B} \sum_{t'=1}^{T_j} \exp(-k \cdot H_{\text{norm},t'}^{(i)})} \tag{6}$$

For a confident step ($H_t^{(i)}$ is lower than the batch average), $g(H_t^{(i)}) > 1$, which **amplifies** its gradient. This accelerates convergence for confident and correct decisions ($A^{(i)} > 0$) and provides a strong corrective penalty for confident errors ($A^{(i)} < 0$), combating "hallucinated confidence". Conversely, for an uncertain step ($H_t^{(i)}$ is higher than average), $g(H_t^{(i)}) < 1$, which **attenuates** its gradient, preventing noisy updates from high-entropy exploration from destabilizing the policy.

---

**Algorithm 1** Entropy-Modulated Policy Gradients (EMPG)

---

1: **Initialize:** Policy $\pi_\theta$.
2: **for** each training iteration **do**
3:     Collect a batch of trajectories $\mathcal{B} = \{\tau_i\}$ by running policy $\pi_\theta$.
4:     Calculate outcome-based advantages $A^{(i)}$ for each trajectory $\tau_i \in \mathcal{B}$.
5:     Compute all step-level entropies $\{H_t\}$ for all steps in the batch.
6:     Normalize all entropies $\{H_t\}$ to $\{H_{\text{norm},t}\}$ using batch min-max scaling.
7:     Compute the self-calibrating scaling factors $\{g(H_t)\}$ for all steps using Eq. 6.
8:     **for** each step $t$ in each trajectory $\tau_i$ **do**
9:         Calculate future clarity bonus $f(H_{t+1}^{(i)})$ using Eq. 7.
10:         Compute modulated advantage $A_{\text{mod}}(i, t)$ using Eq. 4.
11:     **end for**
12:     Normalize the batch of all modulated advantages to get $\{A_{\text{final}}(i, t)\}$.
13:     Update policy parameters $\theta$ using policy gradients with $\{A_{\text{final}}(i, t)\}$.
14: **end for**

---

**Future Clarity Bonus** $f(H)$.    Beyond re-calibrating individual step updates, EMPG also encourages the agent to find globally stable and predictable solution paths. The second term in Eq. 4 serves as an intrinsic motivation for this goal:

$$f(H_{t+1}^{(i)}) = \exp(-k' \cdot H_{\text{norm},t+1}^{(i)}) \tag{7}$$

This term adds a positive bonus proportional to the confidence (low entropy) of the **next** step. Weighted by the hyperparameter $\zeta > 0$, this "future clarity" bonus actively guides the agent away from states of high confusion and towards sequences of high-quality, unambiguous decisions.

### 4.3 NORMALIZATION PROCEDURES

**Batch-Level Entropy Normalization.**    To ensure the modulation function $g(H)$ operates on a consistent scale, we normalize step-level entropies within each training batch using min-max scaling. This stateless approach allows the normalization to adapt dynamically to the policy's evolving confidence level. For each entropy value $H_t$ in the batch:

$$H_{\text{norm},t}^{(i)} = \frac{H_t^i - \min_{\text{batch}}(H)}{\max_{\text{batch}}(H) - \min_{\text{batch}}(H) + \epsilon} \tag{8}$$

**Final Advantage Normalization.**    After computing the modulated advantage $A_{\text{mod}}$ for all steps in a batch, we perform a final batch-level normalization (zero mean). This standard variance reduction technique, which is crucial for stable policy updates, is achieved by subtracting the mean of $A_{\text{mod}}$ over the mini-batch of size $N_B$:

$$A_{\text{final}}(i, t) = A_{\text{mod}}(i, t) - \frac{1}{N_B} \sum_{j=1}^{N_B} \sum_{t_j=1}^{T_j} A_{\text{mod}}(j, t_j) \tag{9}$$

The overall EMPG algorithm is summarized in Algorithm 1, with an implementation provided in the appendix H. Furthermore, we provide a rigorous theoretical derivation for the EMPG update rule in Appendix D.

## 5 EXPERIMENTS

### 5.1 EXPERIMENTAL SETUP

**Tasks and Benchmarks.**    We evaluate our method on three challenging long-horizon agent benchmarks featuring sparse, binary success rewards: WebShop (Yao et al., 2022), a web navigation task requiring complex instruction following; ALFWorld (Shridhar et al., 2021), a text-based environment combining instruction following with common-sense reasoning; and Deep Search Jin et al. (2025), a multi-step information retrieval and synthesis task. For Deep Search, we further categorize the evaluation sets into in-domain (ID) and out-of-domain (OOD) to assess generalization.

**Models and Agent Framework.** Our agent employs the ReAct paradigm (Yao et al., 2023), where the LLM first generates a thought before producing an action. For WebShop and ALFWorld, we use Qwen2.5-1.5B-Instruct (Yang et al., 2024) and Qwen2.5-7B-Instruct to compare our results with existing work. For the more complex Deep Search task, we use the powerful Qwen2.5-32B-Instruct model to conduct in-depth analysis.

**Baselines and Implementation.** We compare EMPG against strong policy gradient baselines: GRPO (Shao et al., 2024) and DAPO (Yu et al., 2025). Our method, EMPG, is implemented as an advantage modulation module that is applied directly on top of these baselines. This allows us to fairly measure the benefits of leveraging intrinsic uncertainty signals. For the WebShop and ALFWorld benchmarks, we based our implementation on the public codebase of GiGPO (Feng et al., 2025) for a fair comparison. For the DeepSearch benchmark, we curated a training dataset of 17k instances by filtering from several sources, including WebWalker (Wu et al., 2025), HotpotQA (Yang et al., 2018), 2WikiMultiHopQA (Ho et al., 2020), NaturalQuestions (Kwiatkowski et al., 2019), and TriviaQA (Joshi et al., 2017).

## 5.2 Main Results

Our comprehensive experiments demonstrate that EMPG yields significant and consistent performance improvements across a diverse range of tasks, baselines, and model scales.

**Performance on ALFWorld and WebShop.** As shown in Table 1, EMPG serves as a robust enhancement to existing policy optimization algorithms. On the Qwen2.5-1.5B model, applying EMPG boosts the average success rate of GRPO on ALFWorld by +8.1 points and DAPO by +7.3 points. This effectiveness scales to the larger Qwen2.5-7B model, where EMPG again improves both baselines on ALFWorld and elevates the DAPO success rate on WebShop to 82.7%. These results confirm that EMPG is highly compatible and provides reliable gains for different RL backbones.

**Performance and Scalability on Deep Search.** To investigate the scalability of our approach on more powerful models and complex retrieval tasks, we evaluated EMPG on the Deep Search benchmark using the Qwen2.5-32B-Instruct model. The results, presented in Table 2, further validate our method. Applying EMPG to the strong DAPO baseline boosts the overall average score from 62.0 to 65.3, a substantial improvement of +3.3 points. This performance gain is notably robust, with EMPG improving the in-domain average by +3.1 points and demonstrating even stronger generalization with a +3.9 point gain on out-of-domain tasks.

Taken together, the results across all three benchmarks confirm that EMPG is a versatile and scalable enhancement for training LLM agents. It consistently improves performance regardless of the underlying RL algorithm, the nature of the task, or the size of the base model, validating our core hypothesis that leveraging intrinsic uncertainty is a powerful tool for learning from sparse rewards.

## 5.3 Analysis

To understand the mechanisms behind EMPG's effectiveness, we conduct a series of in-depth analyses focusing on three key questions: (1) What are the individual contributions of EMPG's core components? (2) How does EMPG affect the learning process over time? (3) Why is a step-level analysis of entropy crucial?

**Ablation Study and Generalization Analysis.** To dissect the contributions of our method's two main components, we perform a detailed ablation study using the results from the Deep Search benchmark, as presented in Table 2. The study reveals a distinct and complementary duality in their roles, which stems from how they shape the policy during training. The *Future Clarity Bonus* acts as a powerful *exploitation* signal during training. By reinforcing known, high-quality decision sequences within the training data, it helps the model master the in-domain distribution, leading to a strong performance gain of *+2.6* points on ID tasks. Conversely, the *Self-Calibrating Gradient Scaling* serves as a powerful *regularization* mechanism during training, teaching the model how to behave when it is uncertain. By attenuating updates for high-entropy steps, it produces a final policy that is inherently more robust and less brittle. This learned robustness is then observed during

Table 1: Performance on ALFWorld and WebShop. Results are averaged over 3 random seeds. For ALFWorld, we report the average success rate (%) for each subtask as well as the overall result. For WebShop, we report both the average score and the average success rate (%). Methods marked with * are our reproduced results. The remaining results are adopted from GiGPO Feng et al. (2025).

| Method | ALFWorld | | | | | | | WebShop | |
|---|---|---|---|---|---|---|---|---|---|
| | Pick | Look | Clean | Heat | Cool | Pick2 | All | Score | Succ. |
| *Base: Closed-Source Model* | | | | | | | | | |
| Prompting GPT-4o | 75.3 | 60.8 | 31.2 | 56.7 | 21.6 | 49.8 | 48.0 | 31.8 | 23.7 |
| Prompting Gemini-2.5-Pro | 92.8 | 63.3 | 62.1 | 69.0 | 26.6 | 58.7 | 60.3 | 42.5 | 35.9 |
| *Base: Qwen2.5-1.5B-Instruct* | | | | | | | | | |
| Prompting Qwen2.5 | 5.9 | 5.5 | 3.3 | 9.7 | 4.2 | 0.0 | 4.1 | 23.1 | 5.2 |
| Prompting ReAct | 17.4 | 20.5 | 15.7 | 6.2 | 7.7 | 2.0 | 12.8 | 40.1 | 11.3 |
| Prompting Reflexion | 35.3 | 22.2 | 21.7 | 13.6 | 19.4 | 3.7 | 21.8 | 55.8 | 21.9 |
| RL Training PPO (with critic) | 64.8 | 40.5 | 57.1 | 60.6 | 46.4 | 47.4 | 54.4 | 73.8 | 51.5 |
| RL Training RLOO | 88.3 | 52.8 | 71.0 | 62.8 | 66.4 | 56.9 | 69.7 | 73.9 | 52.1 |
| RL Training GRPO* | $87.9 \pm 6.3$ | $40.0 \pm 5.8$ | $78.1 \pm 3.8$ | $35.7 \pm 4.3$ | $65.2 \pm 1.2$ | $44.4 \pm 1.4$ | $65.6 \pm 2.9$ | $78.0 \pm 1.1$ | $58.2 \pm 2.4$ |
| with EMPG* | $85.5 \pm 4.8$ | $33.5 \pm 6.4$ | $78.9 \pm 2.5$ | $76.2 \pm 9.7$ | $74.7 \pm 1.9$ | $69.1 \pm 6.4$ | $73.7 \pm 2.7$ (+8.1) | $80.4 \pm 0.7$ | $60.8 \pm 1.3$ (+2.6) |
| RL Training DAPO* | $88.1 \pm 4.7$ | $61.4 \pm 4.4$ | $82.5 \pm 3.4$ | $\mathbf{90.1} \pm 7.3$ | $83.9 \pm 0.8$ | $69.5 \pm 4.9$ | $80.8 \pm 1.4$ | $85.9 \pm 1.3$ | $73.2 \pm 1.3$ |
| with EMPG* | $\mathbf{97.7} \pm 0.8$ | $\mathbf{80.7} \pm 6.9$ | $\mathbf{87.5} \pm 3.2$ | $87.0 \pm 3.6$ | $\mathbf{88.3} \pm 4.1$ | $\mathbf{80.0} \pm 5.6$ | $\mathbf{88.1} \pm 2.1$ (+7.3) | $\mathbf{86.8} \pm 1.9$ | $\mathbf{73.8} \pm 1.1$ (+0.6) |
| *Base: Qwen2.5-7B-Instruct* | | | | | | | | | |
| Prompting Qwen2.5 | 33.4 | 21.6 | 19.3 | 6.9 | 2.8 | 3.2 | 14.8 | 26.4 | 7.8 |
| Prompting ReAct | 48.5 | 35.4 | 34.3 | 13.2 | 18.2 | 17.6 | 31.2 | 46.2 | 19.5 |
| Prompting Reflexion | 62.0 | 41.6 | 44.9 | 30.9 | 36.3 | 23.8 | 42.7 | 58.1 | 28.8 |
| RL Training PPO (with critic) | 92.3 | 64.0 | 92.5 | 89.5 | 80.3 | 68.8 | 80.4 | 81.4 | 68.7 |
| RL Training RLOO | 87.6 | 78.2 | 87.3 | 81.3 | 71.9 | 48.9 | 75.5 | 80.3 | 65.7 |
| RL Training GRPO* | $88.8 \pm 5.6$ | $43.7 \pm 8.2$ | $88.1 \pm 3.5$ | $70.3 \pm 6.9$ | $77.7 \pm 2.3$ | $56.8 \pm 9.4$ | $74.8 \pm 3.1$ | $77.8 \pm 1.4$ | $65.6 \pm 1.0$ |
| with EMPG* | $92.9 \pm 2.9$ | $75.2 \pm 3.8$ | $74.8 \pm 3.9$ | $86.3 \pm 4.7$ | $73.7 \pm 2.6$ | $65.3 \pm 5.8$ | $78.5 \pm 1.7$ (+3.7) | $81.0 \pm 1.4$ | $69.3 \pm 0.5$ (+3.7) |
| RL Training DAPO* | $98.9 \pm 1.4$ | $86.1 \pm 7.1$ | $94.9 \pm 1.6$ | $83.2 \pm 6.4$ | $\mathbf{81.4} \pm 2.6$ | $90.1 \pm 2.2$ | $90.0 \pm 1.1$ | $90.6 \pm 0.5$ | $79.6 \pm 0.6$ |
| with EMPG* | $\mathbf{99.0} \pm 0.3$ | $\mathbf{86.8} \pm 5.5$ | $\mathbf{97.3} \pm 0.9$ | $\mathbf{94.9} \pm 3.9$ | $75.8 \pm 3.4$ | $\mathbf{90.3} \pm 3.1$ | $\mathbf{91.6} \pm 0.8$ (+1.6) | $\mathbf{92.0} \pm 1.2$ | $\mathbf{82.7} \pm 1.0$ (+3.1) |

Table 2: Main results on Deep Search tasks, categorized by domain. EMPG demonstrates strong performance on both in-domain (ID) and out-of-domain (OOD) datasets, with a particularly notable gain in generalization to OOD tasks.

| Method | In-domain (ID) | | | | Out-of-domain (OOD) | | | Overall |
|---|---|---|---|---|---|---|---|---|
| | WebWalker | HotpotQA | 2wiki | Avg. | Musique | Bamboogle | Avg. | Avg. |
| **Qwen2.5-32B-Instruct** | | | | | | | | |
| DAPO (Baseline) | 55.1 | 66.4 | 68.9 | 63.5 | 38.8 | 80.8 | 59.8 | 62.0 |
| *Ablation Studies* | | | | | | | | |
| + Gradient Scaling | 54.9 | 68.8 | 67.4 | 63.7 | 41.0 | 86.4 | 63.7 | 63.7 |
| + Future Bonus | **60.6** | 69.7 | 67.9 | 66.1 | 40.4 | 82.4 | 61.4 | 64.2 |
| **+ EMPG (Ours)** | 57.5 | **71.2** | **71.0** | **66.6** | 41.8 | 84.8 | 63.7 | **65.3** |
| **Gain vs. Baseline** | (+2.4) | (+4.8) | (+2.1) | **(+3.1)** | (+3.0) | (+4.0) | **(+3.9)** | **(+3.3)** |

testing on out-of-domain tasks, where the model faces novel inputs that induce high uncertainty. Because the policy has learned not to overreact in such situations, it exhibits superior generalization, providing a robust gain of *+3.9* points on OOD tasks. This demonstrates that EMPG is not merely overfitting; instead, by learning a fundamental skill of how to handle uncertainty, it acquires a more resilient problem-solving approach that generalizes effectively. Crucially, the full EMPG model, which integrates both mechanisms, demonstrates a powerful synergy: the model learns to efficiently exploit known patterns while being robust to novel ones.

**Enhancing Training Stability.** Beyond improving sample efficiency, EMPG also significantly enhances the stability and robustness of the training process. A common failure mode in online RL fine-tuning is "policy collapse," where the agent's policy diverges late in training, leading to a catastrophic drop in performance. We visualize this phenomenon by tracking the KL Loss during training, as shown in Figure 2. The DAPO baseline agent initially learns effectively, but its KL Loss becomes highly erratic after approximately 240 training steps, indicating severe instability. In contrast, the EMPG-enhanced agent maintains a low and stable KL Loss throughout the entire training run. This demonstrates that EMPG's mechanisms, particularly the self-calibrating gradient scaling, effectively regularize the policy updates, preventing the overly aggressive changes that can lead to divergence and ensuring a more reliable convergence to a high-performance policy. To ensure a fair comparison, **we select the checkpoint at 220 steps for both the baseline and EMPG** for final evaluation. Despite this, our method could continue to improve its performance with further training.

**Step-Level vs. Token-Level Entropy Dynamics.** Our work diverges from prior analyses (Wang et al., 2025) by focusing on entropy at the "reason-act" step level rather than the token level. To

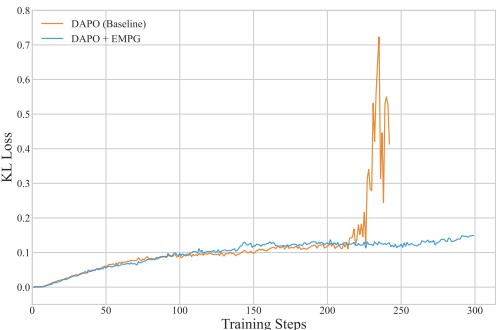

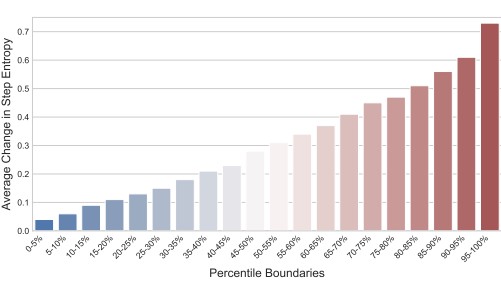

Figure 2: KL Loss dynamics during training for the Qwen2.5-32B-Instruct model. The DAPO baseline (orange) suffers from late-stage instability, evidenced by the sharp, erratic spike in KL Loss. The EMPG-enhanced model (blue) remains stable throughout, showcasing its robustness.

Figure 3: Average entropy change after RL fine-tuning within each 5% entropy percentile range. Unlike token-level findings, even low-entropy steps undergo significant changes, validating our step-level analysis.

validate this choice, we investigate whether the token-level observation—that RL updates primarily affect high-entropy tokens—holds at the step level. We analyze over 9,000 steps on ALFWorld and plot the average entropy change for steps, binned by their initial entropy percentile (Figure 3). Our findings are significant: unlike at the token level, even steps with very low initial entropy (e.g., the 15%-20% percentile) still undergo substantial average entropy changes. This shows the dynamics do not transfer; a confident step can still require significant policy updates. This key finding underscores the importance of our step-centric approach and motivates the design of EMPG to modulate updates across the entire confidence spectrum.

**Analysis of Learning Dynamics.** An analysis of the learning dynamics, presented in Figure F.1, reveals EMPG's critical role in overcoming the performance limitations of baseline methods. Across all experiments on both the ALFWorld and WebShop benchmarks, the baseline agents consistently reach a distinct performance plateau, where their learning stagnates and the success rate ceases to improve. In stark contrast, the EMPG-enhanced agents decisively break through this performance ceiling. By providing a richer and more effective learning signal, EMPG enables the agents to sustain their learning momentum, pushing beyond the baseline's peak and ultimately converging to a significantly higher final success rate. This demonstrates that EMPG is not just accelerating learning, but is fundamentally guiding the agent to discover superior policies that are otherwise inaccessible, effectively escaping the local optima where the baseline methods become trapped.

## 6 CONCLUSION

In this work, we introduced Entropy-Modulated Policy Gradients (EMPG), a novel and principled framework to alleviate the long-standing credit assignment problem in long-horizon LLM agent training. By leveraging the intrinsic uncertainty of the agent's "reasoning-action" steps, EMPG dynamically re-calibrates the policy gradient, moving beyond the limitations of sparse, end-of-task rewards. Our method directly addresses the dual challenges of standard policy gradients: it amplifies updates for confident and correct actions, strongly penalizes confident but incorrect steps, and attenuates updates for uncertain steps to promote stability. Through comprehensive experiments on challenging long-horizon benchmarks, including WebShop, ALFWorld, and Deep Search, we demonstrated substantial performance gains over strong baselines like GRPO and DAPO. More fundamentally, our work addresses a key optimization challenge inherent in policy gradient methods operating over high-dimensional, sequential generative policies (such as Large Language Models): the "entropy-gradient coupling" problem. We frame EMPG as a robust and adaptive policy optimization technique for these agents, designed to dynamically assign credit by utilizing the policy's own intrinsic uncertainty as a reliable, step-level signal.

Our findings suggest that an agent's intrinsic uncertainty is a powerful, yet underexplored, signal for self-supervision in complex decision-making processes. EMPG provides a scalable alternative to costly process-based reward models, forging a dense, informative learning signal from minimal external feedback. For future work, we plan to explore the application of EMPG to other long-horizon tasks, such as embodied AI and multi-agent collaboration. We believe that this work lays a foundational stone for developing more efficient, robust, and self-correcting autonomous agents.

## ETHICS STATEMENT

We confirm that this work adheres to the ICLR Code of Ethics. This research focuses on fundamental algorithms for improving the training efficiency of LLM agents. Our experiments are based entirely on publicly available models and datasets and do not involve any private data. The authors are fully responsible for the content and integrity of this research.

## REPRODUCIBILITY STATEMENT

We are committed to ensuring the full reproducibility of this research. All experiments are based on the publicly available models and public benchmarks. We provide all necessary hyperparameters, computational environments, and pseudocode for the core logic in the appendix of our paper, and key results are reported as averages over multiple random seeds to ensure stability.

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

## A  LLM USAGE DISCLOSURE

We used large language models as a writing assistant to polish and improve the clarity of the English language in this manuscript. The model was not used to generate any core content, including research ideas, experimental results, or technical analysis. All authors have reviewed and are fully responsible for the final content of the paper.

## B  DETAILED PRELIMINARIES

### B.1  POLICY OPTIMIZATION IN REINFORCEMENT LEARNING

Our work is grounded in policy gradient methods, which seek to optimize a policy $\pi_\theta$ parameterized by $\theta$ to maximize the expected reward objective:

$$\mathcal{J}(\pi_\theta) := \mathbb{E}_{\tau \sim \pi_\theta}[R(\tau)] \tag{10}$$

where $\tau$ is a trajectory sampled under policy $\pi_\theta$ and $R(\tau)$ is its total return. The policy gradient theorem allows for direct optimization of this objective via gradient ascent. The gradient is estimated as an expectation over trajectories:

$$\nabla_\theta \mathcal{J}(\pi_\theta) = \mathbb{E}_{\tau \sim \pi_\theta} \left[ \sum_{t=0}^{T} A(s_t, a_t) \nabla_\theta \log \pi_\theta(a_t|s_t) \right] \tag{11}$$

where $s_t$ and $a_t$ are the state and action at time step t, respectively.

A key challenge in estimating this gradient is its inherently high variance. To mitigate this, an advantage function, $A(s_t, a_t)$, is used to measure the relative quality of an action. This advantage is typically estimated using a learned value model, which predicts the expected return from a given state (Schulman et al., 2017). However, this approach has significant drawbacks. The value model is often comparable in size to the policy model, introducing substantial memory and computational overhead. Furthermore, the effectiveness of the algorithm hinges on the reliability of its value estimates, which are inherently difficult to learn accurately Liu et al. (2024); Kazemnejad et al. (2024), especially for complex tasks with long response horizons. Due to these challenges, value-free methods, which estimate the advantage directly from sampled trajectories without a learned value function, have become increasingly popular (Shao et al., 2024; Yu et al., 2025). Our work is also grounded in this value-free paradigm, foregoing a value model to improve training efficiency and stability.

### B.2  RL FRAMEWORK FOR LONG-HORIZON AGENT TASKS

We formalize the long-horizon task as a standard reinforcement learning problem. An LLM agent interacts with an environment over a trajectory $\tau = (s_0, a_0, r_0, ..., s_T, a_T, r_T)$. The reward signal is sparse, with $r_t = 0$ for all non-terminal steps. Assuming an undiscounted setting ($\gamma = 1$), the trajectory return $R(\tau)$ is thus determined solely by the final outcome:

$$R(\tau) = \sum_{t=0}^{T} \gamma^t r_t = r_T \in \{0, 1\} \tag{12}$$

In our work, a single step corresponds to a complete "reason-then-act" cycle (e.g., as in ReAct (Yao et al., 2023)), forming a multi-step decision-making process. This sparse-reward, long-horizon setting epitomizes two fundamental RL challenges: the **credit assignment problem** and the **exploration problem**.

### B.3  STRATEGIES FOR LEARNING FROM SPARSE OUTCOME-BASED REWARDS

To enable effective learning from sparse, outcome-based rewards in long-horizon tasks, several powerful strategies have emerged that form the foundation of modern LLM RL.

- **Trust Region Learning.** Proximal Policy Optimization (PPO) (Schulman et al., 2017) serves as the bedrock algorithm. Its primary innovation is not credit assignment, but ensuring

training stability. It achieves this by constraining policy updates within a trust region, using a clipped objective on the probability ratio $\rho_t(\theta) = \frac{\pi_\theta(a_t|s_t)}{\pi_{\theta_{\text{old}}}(a_t|s_t)}$. When applied to sparse reward tasks, PPO's effectiveness fundamentally depends on the quality of its advantage estimates, which implicitly perform the task of credit assignment Kazemnejad et al. (2024).

- **Group-Based Advantage Estimation.** Group Relative Policy Optimization (GRPO) (Shao et al., 2024) builds upon this foundation with a direct solution for credit assignment. It addresses the high variance of the policy gradient inherent in sparse rewards by sampling multiple responses ($M$) and computing a Z-score-like advantage:

$$A_{ij} = \frac{r(x_i, y_{ij}) - \text{mean}_{k=1}^M(r(x_i, y_{ik}))}{\text{std}_{k=1}^M(r(x_i, y_{ik})) + \epsilon} \tag{13}$$

  Here, $r(x_i, y_{ij})$ is the final outcome-based reward for the j-th response, and $\epsilon$ is a small constant added for numerical stability. This comparative evaluation effectively identifies the best-in-batch responses, providing a robust signal.

- **Adaptive Data Curation.** Decoupled Clip and Dynamic Sampling Policy Optimization (DAPO) (Yu et al., 2025) further refines the learning process by curating the data itself. It addresses failure modes in GRPO by filtering and resampling trajectories to form more informative training batches. By focusing updates on a buffer of high-quality samples, it improves the efficiency of learning from the sparse reward signal.

While powerful, these strategies share a common reliance on processing external, outcome-based reward signals. As they are primarily designed for single-turn generation, they treat entire action sequences as monolithic blocks. When applied to interactive agent tasks, this leads to a coarse, trajectory-level credit assignment that fails to pinpoint which specific actions in a long sequence were critical for success. This approach ignores the rich, intrinsic signals available at each step of the generative process. Our work diverges by proposing a new paradigm that peers inside the model, leveraging its intrinsic, step-wise uncertainty.

## C  PROOF OF PROPOSITION 1

We aim to prove that $\mathbb{E}_{a_k \sim \pi}\left[||\nabla_z \log \pi_k||^2\right] = 1 - \sum_{j=1}^{|V|} \pi_j^2$. The proof requires the result for the gradient norm of a single action $a_k$, which we state as a lemma.

**Lemma.** The squared L2-norm of the score function with respect to the logits, for a chosen action $a_k$, is given by: $||\nabla_z \log \pi_k||^2 = 1 - 2\pi_k + \sum_{j=1}^{|V|} \pi_j^2$.

*Proof of Lemma.* Let the logits be $z = (z_1, \ldots, z_{|V|})$. The policy is $\pi_k = \exp(z_k)/\sum_j \exp(z_j)$. The partial derivative of the log-probability $\log \pi_k$ with respect to an arbitrary logit $z_i$ is $\frac{\partial \log \pi_k}{\partial z_i} = \delta_{ik} - \pi_i$, where $\delta_{ik}$ is the Kronecker delta. The squared L2-norm of the gradient vector $\nabla_z \log \pi_k$ is therefore:

$$||\nabla_z \log \pi_k||^2 = \sum_{i=1}^{|V|} (\delta_{ik} - \pi_i)^2 = (1 - \pi_k)^2 + \sum_{i \neq k} (-\pi_i)^2$$

$$= (1 - 2\pi_k + \pi_k^2) + \sum_{i \neq k} \pi_i^2 = 1 - 2\pi_k + \sum_{j=1}^{|V|} \pi_j^2$$

∎

*Proof of Proposition 1.* The expectation is taken over all possible choices of action $a_k$ according to the policy distribution $\pi$. Using the result from the lemma:

$$
\begin{aligned}
\mathbb{E}_{k \sim \pi}\left[||\nabla_z \log \pi_k||^2\right] &= \sum_{k=1}^{|V|} \pi_k \cdot \left(||\nabla_z \log \pi_k||^2\right) \\
&= \sum_{k=1}^{|V|} \pi_k \left(1 - 2\pi_k + \sum_{j=1}^{|V|} \pi_j^2\right) \\
&= \sum_{k=1}^{|V|} \pi_k - 2\sum_{k=1}^{|V|} \pi_k^2 + \sum_{k=1}^{|V|} \pi_k \left(\sum_{j=1}^{|V|} \pi_j^2\right) \\
&= 1 - 2\sum_{k=1}^{|V|} \pi_k^2 + \left(\sum_{j=1}^{|V|} \pi_j^2\right)\left(\sum_{k=1}^{|V|} \pi_k\right) \quad \text{(Factor out constant term)} \\
&= 1 - 2\sum_{k=1}^{|V|} \pi_k^2 + \left(\sum_{j=1}^{|V|} \pi_j^2\right) \cdot 1 \\
&= 1 - \sum_{k=1}^{|V|} \pi_k^2
\end{aligned}
$$

Recalling the definition of Rényi entropy of order 2, $H_2(\pi) = -\log(\sum_{j=1}^{|V|} \pi_j^2)$, we can identify the term $\sum \pi_j^2$ as the collision probability, which is equivalent to $\exp(-H_2(\pi))$. Substituting this into our result yields the final information-theoretic form:

$$
\mathbb{E}_{k \sim \pi}\left[||\nabla_z \log \pi_k||^2\right] = 1 - \exp(-H_2(\pi))
$$

This completes the proof of the proposition. ∎

## D  THEORETICAL FOUNDATION OF THE EMPG UPDATE RULE

In this section, we provide a rigorous theoretical justification for the Entropy-Modulated Policy Gradients (EMPG) algorithm. We demonstrate that the EMPG update rule can be formally derived as the gradient of a composite objective function, $J_{\text{EMPG}}(\theta)$. This interpretation substantiates that EMPG is a principled optimization method that reshapes the standard reinforcement learning objective to favor policies that are both effective and robust.

### D.1  THE STANDARD POLICY GRADIENT OBJECTIVE

We begin with the standard objective in policy-based reinforcement learning, which is to maximize the expected total return. In the context of sparse, outcome-based rewards, this objective simplifies to maximizing the expected advantage (return) of a trajectory $\tau$:

$$
J(\theta) = \mathbb{E}_{\tau \sim \pi_\theta}[A^{(\tau)}] \tag{14}
$$

where $A^{(\tau)}$ is the scalar return for a trajectory $\tau$ sampled from the policy $\pi_\theta$. The gradient of this objective is given by the Policy Gradient Theorem:

$$
\nabla_\theta J(\theta) = \mathbb{E}_{\tau \sim \pi_\theta}\left[\left(\sum_{t=0}^{T-1} \nabla_\theta \log \pi_\theta(a_t|s_t)\right) A^{(\tau)}\right] \tag{15}
$$

For any single trajectory $\tau$, the gradient estimator is $\mathcal{G}^{(\tau)}(\theta) = A^{(\tau)} \sum_{t=0}^{T-1} \nabla_\theta \log \pi_\theta(a_t|s_t)$. This formulation reveals the core issue identified in Proposition 1: the contribution of each step's score function, $\nabla_\theta \log \pi_\theta(a_t|s_t)$, is weighted uniformly by the trajectory's outcome $A^{(\tau)}$, while its norm is intrinsically coupled with the policy entropy $H_t$.

## D.2 THE EMPG COMPOSITE OBJECTIVE FUNCTION

We posit that EMPG performs gradient ascent on a composite objective function $J_{\text{EMPG}}(\theta)$. This objective augments the standard RL objective with a term that explicitly accounts for policy uncertainty, thereby decoupling the learning signal's magnitude and direction from the policy's raw confidence. We define this objective as:

$$J_{\text{EMPG}}(\theta) = J_{\text{extrinsic}}(\theta) + J_{\text{intrinsic}}(\theta) \tag{16}$$

Here, $J_{\text{extrinsic}}(\theta)$ is a re-weighted extrinsic objective that addresses the gradient *magnitude* problem, and $J_{\text{intrinsic}}(\theta)$ is an intrinsic objective that guides the policy's *direction* towards states of higher certainty.

### D.2.1 THE RE-WEIGHTED EXTRINSIC OBJECTIVE

The self-calibrating gradient scaling component of EMPG, $A^{(\tau)} \cdot \mathcal{H}$, can be interpreted as performing an update on a modified extrinsic objective. Formally, we define a state-dependent weighting function $\omega(s_t, \theta) = \mathcal{H}$, which is a function of the policy's entropy at state $s_t$. The gradient update for this component is:

$$\mathcal{G}_{\text{extrinsic}}^{(\tau)}(\theta) = \sum_{t=0}^{T-1} A^{(\tau)} \cdot \omega(s_t^{(\tau)}, \theta) \cdot \nabla_\theta \log \pi_\theta(a_t|s_t) \tag{17}$$

This formulation is equivalent to optimizing the standard objective $J(\theta)$ under a *state-dependent measure*, where the contribution of each state is re-weighted. While deriving a closed-form objective $J_{\text{extrinsic}}(\theta)$ is non-trivial because $\omega$ depends on $\theta$ in a complex manner (via batch statistics), this interpretation is sufficient to justify the update rule. The weighting function $\omega(s_t, \theta)$ serves as an adaptive, information-theoretic learning rate that directly counteracts the dynamics described in Proposition 1. It amplifies the learning signal for confident (low-entropy) steps and dampens it for uncertain (high-entropy) steps, thus achieving a direct re-calibration of the gradient's magnitude.

### D.2.2 THE INTRINSIC CLARITY OBJECTIVE

The Future Clarity Bonus can be modeled as the gradient of a well-defined intrinsic objective function. We define an intrinsic reward, $r_t^{\text{int}}$, awarded at step $t$ for transitioning to a state $s_{t+1}$ with high policy clarity:

**Definition** (Clarity Reward). The intrinsic clarity reward at step $t$ is a function of the policy entropy at the subsequent state $s_{t+1}$:

$$r_t^{\text{int}}(s_{t+1}; \theta) = \zeta \cdot f(H(\pi_\theta(\cdot|s_{t+1}))) = \zeta \cdot \exp(-k' \cdot H_{\text{norm},t+1}) \tag{18}$$

This reward incentivizes actions that lead to predictable future states. The corresponding intrinsic objective, $J_{\text{intrinsic}}(\theta)$, is the expected cumulative intrinsic reward:

$$J_{\text{intrinsic}}(\theta) = \mathbb{E}_{\tau \sim \pi_\theta} \left[ \sum_{t=0}^{T-1} r_t^{\text{int}}(s_{t+1}; \theta) \right] \tag{19}$$

Applying the policy gradient theorem to this objective, and using the immediate intrinsic reward as a one-step advantage estimate (a common form of advantage shaping), yields the gradient:

$$\nabla_\theta J_{\text{intrinsic}}(\theta) = \mathbb{E}_{\tau \sim \pi_\theta} \left[ \sum_{t=0}^{T-1} (\nabla_\theta \log \pi_\theta(a_t|s_t)) \, r_t^{\text{int}}(s_{t+1}; \theta) \right] \tag{20}$$

$$= \mathbb{E}_{\tau_i \sim \pi_\theta} \left[ \sum_{t=0}^{T_i-1} (\nabla_\theta \log \pi_\theta(a_t|s_t)) \zeta \cdot f(H_{t+1}^{(\tau)}) \right] \tag{21}$$

This gradient precisely matches the Future Clarity Bonus component of the EMPG update.

### D.3 Synthesis: The Full EMPG Gradient

By combining the gradients of the extrinsic and intrinsic objectives, we recover the full EMPG gradient estimator for a single trajectory $\tau$:

$$\mathcal{G}_{\text{EMPG}}^{(\tau)}(\theta) = \mathcal{G}_{\text{extrinsic}}^{(\tau)}(\theta) + \nabla_\theta J_{\text{intrinsic}}(\theta)|_\tau \tag{22}$$

$$= \sum_{t=0}^{T-1} A^{(\tau)} \cdot \mathcal{H} \cdot \nabla_\theta \log \pi_\theta(a_t|s_t) + \sum_{t=0}^{T-1} \zeta \cdot f(H_{t+1}^{(\tau)}) \cdot \nabla_\theta \log \pi_\theta(a_t|s_t) \tag{23}$$

$$= \sum_{t=0}^{T-1} \left( A^{(\tau)} \cdot \mathcal{H} + \zeta \cdot f(H_{t+1}^{(\tau)}) \right) \nabla_\theta \log \pi_\theta(a_t|s_t) \tag{24}$$

This derivation confirms that the EMPG algorithm performs a principled gradient ascent on the composite objective $J_{\text{EMPG}}(\theta)$. This objective function holistically reshapes the optimization landscape by (1) adaptively scaling the extrinsic reward signal to ensure its magnitude is motivationally salient rather than merely a function of policy entropy, and (2) introducing an intrinsic drive towards robust, predictable solution paths. This dual-pronged approach provides a theoretical foundation for why EMPG successfully mitigates the challenges posed by the inherent dynamics of standard policy gradients.

## E Experimental Settings

This appendix provides a detailed description of the experimental settings, hardware configurations, and hyperparameter choices for our experiments across the three main benchmarks. Due to the differences in training frameworks and task environments, the settings for WebShop/ALFWorld and Deep Search are described in separate subsections.

### E.1 WebShop and ALFWorld Experiments

Our experiments on WebShop and ALFWorld are conducted within the **Verl-Agent** framework, an extension of the **veRL** Sheng et al. (2024) training codebase specifically designed for training large language model (LLM) agents via reinforcement learning. Verl-Agent provides a powerful and scalable platform for long-horizon, multi-turn RL training by enabling fully customizable per-step input structures, history management, and memory modules. It supports a diverse set of RL algorithms and a rich suite of agent environments, making it highly suitable for our work.

For a fair comparison, all experiments were re-executed on our hardware platform. While the original experiments were performed using H200 GPUs, our work utilized A100 GPUs due to resource constraints. We observed that the original training scripts for the Qwen2.5-1.5B-Instruct model, designed for $2 \times$ H100, would result in out-of-memory errors on A100s. Therefore, we used $4 \times$ A100 GPUs for the 1.5B models and $8 \times$ A100 GPUs for the 7B models. All baselines were re-trained under the same hardware, seeds, and settings to ensure strict comparability. The key hyperparameters for these experiments are summarized in Table 3.

### E.2 Deep Search Experiments

Our experiments on the Deep Search task were conducted using an in-house RL training framework. The agent was equipped with two primary tools: Bing Search as the search engine and a web viewer tool capable of reading web page content and summarizing long articles.

A key part of the Deep Search training was the data curation process. We constructed a unique training dataset of 17,000 instances by filtering from a variety of public benchmarks, including WebWalker Wu et al. (2025), HotpotQA Yang et al. (2018), 2WikiMultiHopQA Ho et al. (2020), NaturalQuestions Kwiatkowski et al. (2019), and TriviaQA Joshi et al. (2017). We gratefully acknowledge the initial data collection and preliminary filtering by the DeepResearcher team Zheng et al. (2025). We performed two deeper filtering steps:

1. **Direct Answer Filtering:** We sampled 5 results per question using Doubao-Seed-1.6 (Thinking) Seed et al. (2025). We then filtered out all questions that could be answered

Table 3: Key Hyperparameters for WebShop and ALFWorld Experiments.

| Parameter | Value |
|---|---|
| Actor Learning Rate | 1e-6 |
| KL Loss Coefficient | 0.01 |
| KL Penalty | low var kl |
| Entropy Coefficient | 0.001 |
| Clip High (DAPO) | 0.28 |
| Clip Low (DAPO) | 0.2 |
| Clip Low/High (GRPO) | 0.2 |
| Batch Size | 16 |
| Training Step | 150 |
| Rollout Group Size | 8 |
| Rollout Temperature | 1.0 |
| $\zeta$ | 0.05 |
| $k, k'$ | 1.0 |
| Max Actions (ALFWorld) | 50 |
| Max Actions (WebShop) | 15 |
| History Observation | 2 |
| GPUs | $4 \times$ A100 (1.5B), $8 \times$ A100 (7B) |

directly (where at least one of the 5 results was correct) to ensure the agent learns to use its search tools rather than relying on memorized answers.

2. **Agent Workflow Filtering:** We further filtered the dataset by sampling 8 results using a search workflow built on Doubao-Seed-1.6 (Thinking). We removed data points that were "stably all-correct" to focus the RL training on more challenging instances and improve training efficiency.

The key hyperparameters for the RL training on the Deep Search task are detailed in Table 4.

Table 4: Key Hyperparameters for Deep Search Experiments.

| Parameter | Value |
|---|---|
| Actor Learning Rate | 1e-6 |
| KL Loss Coefficient | 0.001 |
| KL Penalty | low var kl |
| Entropy Coefficient | 0.0 |
| Clip High | 0.28 |
| Clip Low | 0.2 |
| Batch Size | 64 |
| Training Step | 220 |
| Rollout Group Size | 16 |
| Rollout Temperature | 1.0 |
| $\zeta$ | 0.1 |
| $k, k'$ | 1.0 |
| Max Actions | 15 |
| GPUs | $32 \times$ A100 |

## F ANALYSIS OF LEARNING DYNAMICS

This section provides a detailed visualization of the learning dynamics, complementing the analysis in the main body of the paper. Figure F.1 illustrates the training progress of EMPG-enhanced agents compared to their baseline counterparts (GRPO and DAPO) on both the WebShop and ALFWorld benchmarks. As shown in the learning curves, the baseline agents consistently hit a performance

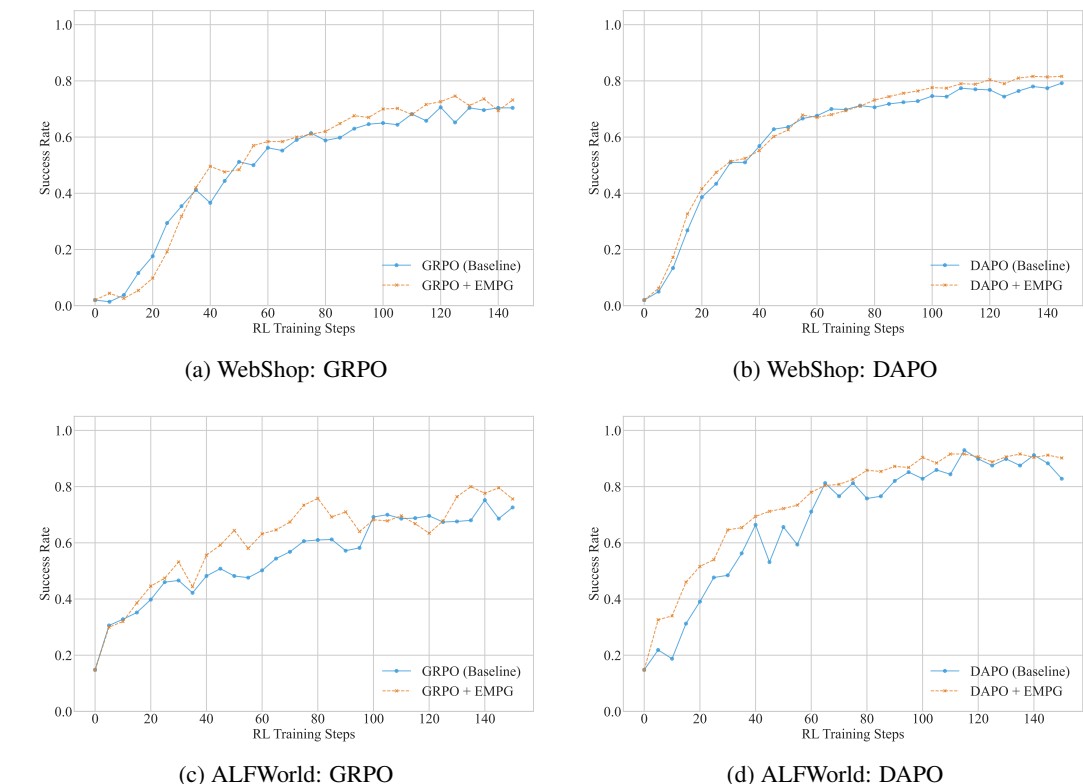

Figure F.1: Learning dynamics comparison for the Qwen2.5-7B-Instruct model on the WebShop and ALFWorld benchmarks (evaluated on the validation set). In all four scenarios, the EMPG-enhanced agents (orange, dashed) demonstrate a superior success rate compared to their respective baselines (blue, solid).

ceiling, with their success rates stagnating early in the training process. In contrast, our EMPG-enhanced agents overcome this plateau, sustaining their learning momentum to achieve significantly higher final success rates across all settings. This evidence supports our central claim that EMPG provides a more effective learning signal, enabling agents to escape the local optima that trap standard policy gradient methods.

# G EXPERIMENTAL ANALYSIS FOR ROBUSTNESS

## G.1 HYPERPARAMETER SENSITIVITY ANALYSIS

We conducted a thorough sensitivity analysis to ensure the practical robustness of EMPG across its primary hyperparameter settings: the Future Clarity Bonus weight ($\zeta$), and the gradient temperature parameters ($k$ and $k'$). These experiments were performed on the ALFWorld benchmark using the Qwen2.5-1.5B-Instruct policy trained with GRPO. For simplicity, we set $k = k'$ as both parameters operate on the same normalized step-level entropy, and we report the average success rate over 4 independent runs for each configuration.

**Analysis of Hyperparameter Sensitivity.** The results in Table 5 confirm that EMPG maintains **stable and competitive performance** across the tested range of hyperparameters ($\zeta \in [0.01, 0.1]$ and $k, k' \in [0.5, 1.5]$).

- **Impact of $k$ and $k'$:** Varying the temperature $k$ (e.g., from 0.5 to 1.5) results in minimal fluctuation in overall performance (73.1% to 73.8%). This stability is a direct and expected benefit of our *Self-Calibrating Gradient Scaling* design (Eq. 6). Since the scaling is normal-

Table 5: Hyperparameter Sensitivity Analysis of EMPG on ALFWorld (Qwen2.5-1.5B-Instruct + GRPO). The method shows strong robustness to variations in $\zeta$ and $k$.

| Parameters | | ALFWorld Subtask Success Rate (%) | | | | | | All (%) |
|---|---|---|---|---|---|---|---|---|
| $\zeta$ | $k, k'$ | Pick | Look | Clean | Heat | Cool | Pick2 | Overall |
| 0.01 | | $83.7 \pm 1.2$ | $68.7 \pm 5.3$ | $78.8 \pm 4.0$ | $72.4 \pm 6.8$ | $59.7 \pm 4.5$ | $69.4 \pm 1.9$ | $73.4 \pm 1.9$ |
| 0.05 | 1.0 | $85.5 \pm 4.8$ | $33.5 \pm 6.4$ | $78.9 \pm 2.5$ | $76.2 \pm 9.7$ | $74.7 \pm 1.9$ | $69.1 \pm 6.4$ | $73.7 \pm 2.7$ |
| 0.1 | | $84.5 \pm 2.0$ | $51.0 \pm 16.2$ | $82.4 \pm 2.1$ | $82.2 \pm 4.5$ | $78.0 \pm 4.3$ | $78.5 \pm 2.7$ | $78.5 \pm 1.2$ |
| | 0.5 | $82.7 \pm 1.5$ | $51.1 \pm 3.1$ | $77.8 \pm 1.9$ | $74.8 \pm 3.9$ | $68.2 \pm 5.5$ | $70.2 \pm 1.5$ | $73.1 \pm 1.3$ |
| 0.05 | 1.0 | $85.5 \pm 4.8$ | $33.5 \pm 6.4$ | $78.9 \pm 2.5$ | $76.2 \pm 9.7$ | $74.7 \pm 1.9$ | $69.1 \pm 6.4$ | $73.7 \pm 2.7$ |
| | 1.5 | $79.5 \pm 2.6$ | $49.9 \pm 3.7$ | $78.1 \pm 1.2$ | $92.0 \pm 1.7$ | $74.0 \pm 4.5$ | $61.4 \pm 7.4$ | $73.8 \pm 1.1$ |

Table 6: Performance on ALFWorld and WebShop. Results are averaged over 3 random seeds. For ALFWorld, we report the average success rate (%) for each subtask as well as the overall result. For WebShop, we report both the average score and the average success rate (%).

| Method | ALFWorld | | | | | | | WebShop | |
|---|---|---|---|---|---|---|---|---|---|
| | Pick | Look | Clean | Heat | Cool | Pick2 | All | Score | Succ. |
| *Base: LLaMA3.1-8B-Instruct* | | | | | | | | | |
| RL Training GRPO | $92.2 \pm 1.7$ | $63.5 \pm 9.5$ | $79.5 \pm 4.7$ | $86.9 \pm 2.0$ | $68.0 \pm 3.3$ | $78.5 \pm 2.8$ | $79.6 \pm 1.7$ | $85.2 \pm 1.4$ | $68.0 \pm 1.0$ |
| with EMPG | $96.8 \pm 1.7$ | $81.2 \pm 3.7$ | $93.3 \pm 1.9$ | $82.6 \pm 3.7$ | $79.5 \pm 3.2$ | $82.9 \pm 2.4$ | $\mathbf{87.5 \pm 1.3}$ | $86.3 \pm 1.2$ | $\mathbf{70.1 \pm 1.5}$ |
| *Base: Qwen2.5-7B-Instruct* | | | | | | | | | |
| RL Training GRPO | $88.8 \pm 5.6$ | $43.7 \pm 8.2$ | $88.1 \pm 3.5$ | $70.3 \pm 6.9$ | $77.7 \pm 2.3$ | $56.8 \pm 9.4$ | $74.8 \pm 3.1$ | $77.8 \pm 1.4$ | $65.6 \pm 1.0$ |
| with EMPG | $92.9 \pm 2.9$ | $75.2 \pm 3.8$ | $74.8 \pm 3.9$ | $86.3 \pm 4.7$ | $73.7 \pm 2.6$ | $65.3 \pm 5.8$ | $\mathbf{78.5 \pm 1.7}$ | $81.0 \pm 1.4$ | $\mathbf{69.3 \pm 0.5}$ |

ized by the mini-batch mean, the mechanism effectively relies on the *relative confidence ranking* within the batch, making the system robust against the absolute value of $k$.

- **Impact of $\zeta$:** The weight of the Future Clarity Bonus ($\zeta$) shows a **clear positive correlation** with the overall success rate, rising from $73.4\%$ at $\zeta = 0.01$ to $78.5\%$ at $\zeta = 0.1$. This trend is highly desirable and reinforces our key theoretical contribution: the Future Clarity Bonus acts as an effective intrinsic signal that guides the agent toward states that lead to clearer, more deterministic future paths. The increasing performance with higher $\zeta$ confirms its intended function as a beneficial, goal-aligned exploitation signal for sequential decision-making.

## G.2 GENERALIZATION AND MODEL ROBUSTNESS

To demonstrate that EMPG is not architecture-specific, we evaluated its effectiveness on the **LLaMA3.1-8B-Instruct** model, a strong baseline from a different model family. We compared the performance of the Baseline (LLaMA3.1-8B-Instruct + GRPO) against the EMPG-enhanced version.

**Empirical Proof of Generalization.** The results in Table 6 confirm that EMPG is a transferable and effective optimization technique. When applied to the LLaMA3.1 architecture, EMPG achieves a significant and consistent uplift on both ALFWorld and WebShop benchmarks. This successfully demonstrates that EMPG's core mechanisms are generalizable across different state-of-the-art Large Language Models.

## H ALGORITHM IMPLEMENTATION DETAILS

We provide a PyTorch-style pseudocode implementation for the core logic of our method in Algorithms 2 and 3. This function calculates the final modulated advantage, $A_{\text{final}}$, used for the policy update, as detailed in Section 4. The process consists of four main stages:

1. **Step-Level Entropy Collection:** The function first iterates through the batch of trajectories to identify agent action steps (i.e., the "assistant" responses). For each step $t$, it computes the corresponding step-level entropy $H_t$ by averaging the policy's token-level entropies for that action.

**Algorithm 2** Part 1: PyTorch-Style Pseudocode for EMPG Advantage Calculation

```python
import numpy as np
import torch

def compute_empg_advantage(tokenizer, batch, k=1.0, k_f=1.0, zeta=0.1):
    """
    Args:
        tokenizer: The tokenizer for identifying response segments.
        batch: A data batch with 'responses', 'old_entropy', 'advantages'.
        k (float): Hyperparameter for self-calibrating gradient scaling.
        k_f (float): Hyperparameter for the future clarity bonus.
        zeta (float): Hyperparameter for the future clarity bonus.
    """
    # --- 1. First Pass: Collect Step-Level Entropies ---
    all_step_entropies = []
    # segments_to_modify stores {'sample_idx', 'start', 'end'} for each
    step
    segments_to_modify = []

    for i in range(batch.batch.batch_size[0]):
        # Find "assistant" segments, which correspond to agent steps.
        token_segments = process_token_sequences(
            batch.batch['responses'][i],
            tokenizer.encode("<|im_start|>assistant\n"),
            tokenizer.encode('<|im_end|>')
        )
        for start, end in token_segments:
            if start >= end: continue

            # Calculate the average token-level entropy for the step
            step_entropy = batch.batch['old_entropy'][i][start:end].mean().item()
            all_step_entropies.append(step_entropy)
            segments_to_modify.append({'sample_idx': i, 'start': start,
'end': end})

    if not all_step_entropies: return
```

2. **Modulation Component Calculation:** All collected step entropies $\{H_t\}$ are normalized across the batch using min-max scaling to produce $\{H_{\text{norm},t}\}$ (as per Eq. 8). These normalized values are then used to compute the two key components of our method: the self-calibrating scaling factor $g(H_t)$ (Eq. 6) and the future clarity bonus term $g'(H_{t+1})$ (Eq. 7).

3. **Advantage Modulation:** The function then applies these components to the original outcome-based advantage. For each step, the advantage is scaled by $g(H_t)$ and augmented by the future clarity bonus $\zeta \cdot g'(H_{t+1})$, yielding the modulated advantage $A_{\text{mod}}$ as defined in our main formula (Eq. 4).

4. **Final Normalization:** Finally, to reduce variance and ensure stable training, the entire batch of resulting modulated advantages is normalized to have a mean of zero. This produces the final advantage $A_{\text{final}}$ (Eq. 9) that is used to compute the policy gradient.

**Algorithm 3** Part 2: PyTorch-Style Pseudocode for EMPG Advantage Calculation (cont.)

```python
# --- 2. Calculate Modulated Advantage Components ---
H = np.array(all_step_entropies)

# Batch-level entropy normalization (Eq. 12) with \epsilon = 1e-8
min_H, max_H = np.min(H), np.max(H)
H_norm = (H - min_H) / (max_H - min_H + 1e-8)

# Self-calibrating gradient scaling g(H) (Eq. 10)
g_H_unnormalized = np.exp(-k * H_norm)
mean_g_H = np.mean(g_H_unnormalized)
g_H = g_H_unnormalized / (mean_g_H + 1e-8)

# Future clarity bonus f(H) (Eq. 11)
f_H = np.exp(-k_f * H_norm)

# Convert to tensors for PyTorch operations
g_H = torch.tensor(g_H, device=batch.batch['advantages'].device,
dtype=torch.float32)
f_H = torch.tensor(f_H, device=batch.batch['advantages'].device,
dtype=torch.float32)

# --- 3. Second Pass: Apply Advantage Modulation (Eq. 8) ---
step_advantages = []
for i, segment in enumerate(segments_to_modify):
    idx, start, end = segment['sample_idx'], segment['start'],
segment['end']

    # Apply self-calibrating gradient scaling
    batch.batch['advantages'][idx][start:end] *= g_H[i]

    # Add future clarity bonus if there is a next step
    next_seg = segments_to_modify[i+1] if i+1 < len(
segments_to_modify) else None
    if next_seg and next_seg['sample_idx'] == idx:
        batch.batch['advantages'][idx][start:end] += zeta * f_H[i+1]
    step_advantages.append(batch.batch['advantages'][idx][start])

# --- 4. Final Advantage Normalization (Eq. 7) ---
if step_advantages:
    final_adv_mean = torch.mean(torch.stack(step_advantages))
    batch.batch['advantages'] -= final_adv_mean
```

