# OpenReview forum: "Harnessing Uncertainty: Entropy-Modulated Policy Gradients for Long-Horizon LLM Agents"
_ICLR.cc/2026/Conference — Submitted to ICLR 2026_

### Official Review · Reviewer_mRR4 · 2025-10-31

**Soundness:** 3
**Presentation:** 3
**Contribution:** 3
**Rating:** 6
**Confidence:** 3

**Summary:**

This paper identifies that magnitude of policy gradients is inherently coupled with the policy’s entropy, and proposed EMPG to recalibrates the learning signal via self-claibrating gradient scaling and future clarity bonus. Experiments on WebShop, ALFWorld, and Deep Search show consistent improvements over GRPO and DAPO.

**Strengths:**

1. The paper formalizes the coupling between entropy and gradient magnitude and propose a correction mechanism rather than heuristic reward shaping.
2. Evaluations span three diverse benchmarks.

**Weaknesses:**

1. Why should the natural coupling between entropy and gradient magnitude be considered harmful rather than a desirable exploration mechanism? Could attenuating high-entropy gradients hinder the discovery of new behaviors? I understand that EMPG is actually anti-uncontrolled entropy coupling, but the proposed entropy-dependent gradient scaling introduces a biased learning signal, and may favor overly confident, low-entropy behaviors and converge to locally optimal or sub-optimal deterministic policies. It will be helpful to clarify whether the method prematurely collapses to confident but non-optimal trajectories.
2. Only three random seeds are reported.

**Questions:**

1. Why is Renyi-2 entropy chosen instead of Shannon entropy, which is more standard in RL formulations?
2. About the scaling function, how sensitive is the method to the hyperparameter $k$, any insight behind the chosen eponential form $e^{-kH_t}$?
3. How sensitive are the results to  $k$, $k'$ and $\zeta$?
4. Appendix D interprets EMPG as optimizing a modified objective $J_{EMPG}$. Is there a guarantee that $J_{EMPG}$ has the same optimal policy as the original $J(\pi)$, if not, what are the advantages and properties of the surrogate objective?
5. Why is step-level entropy superior to token-level entropy (Fig. 3)?
6. How expensive is step-wise entropy computation for large-scale LLMs? And an open question on future directions, can EMPG scale to online or streaming RL settings without prohibitive cost?

---

> ### Author Response · Authors · 2025-11-20
>
> `Q1: Clarifying the Harmful Entropy-Gradient Coupling and Mitigating Premature Collapse.`
>
> We thank the reviewer for this critical question. We understand the perspective that large gradients may drive useful exploration. However, in the context of high-dimensional generative policies (LLMs), we argue that the natural coupling between high entropy and large gradient magnitude becomes detrimental—it primarily introduces noise and instability, rather than effective exploration.
>
> **1. Why the Natural Coupling is Harmful (Noise vs. Exploration)**
>
> The detrimental nature of this coupling stems from the properties of the LLM policy space in long-horizon tasks:
> - Instability in High Dimensions: In the vast, sparse policy space of LLMs, the large L2-norm of the gradient from high-entropy steps (Proposition 1) rarely translates into meaningful exploration. Instead, it introduces excessive noise and high variance, making the policy highly susceptible to catastrophic forgetting or sudden collapse (Figure 2, Baseline failure).
> - The Need for Stability: For multi-step agents requiring stable credit assignment across long, sparse trajectories, stability is paramount. Uncontrolled high-variance updates fundamentally corrupt the learning signal, making the overall policy unusable.
> - By stabilizing these noisy updates, EMPG's Self-Calibrating Gradient Scaling ensures the agent can execute necessary exploration safely without diverging, allowing the entire system to survive long enough to find solutions.
>
> **2. Countering the Risk of Premature Collapse**
>
> The concern that EMPG might favor low-entropy behaviors and prematurely collapse to sub-optimal policies is refuted by our framework’s design and empirical evidence:
> - Future Clarity Bonus Prevents Sub-optimality: The Future Clarity Bonus ensures that low-entropy behaviors are goal-aligned by rewarding actions that lead to predictable future states (solvability), preventing collapse into confident but non-optimal trajectories.
> - Empirical Proof (Anti-Collapse): EMPG achieves the highest overall success rate on multiple benchmarks (Table 1 and Table 2), demonstrating convergence to a superior optimum. Additionally, Figure 2 shows EMPG maintains the stable performance throughout training, directly confirming our method ensures robust and sustained learning rather than premature collapse.
>
> In conclusion, EMPG’s biased gradient scaling is a necessary regulatory mechanism that mitigates the noise inherent in LLM policy optimization, prioritizing stability to achieve superior long-term performance.
>
> `Q2: Rationale for Using Rényi-2 Entropy in Theory vs. Shannon Entropy in Practice.`
>
> We thank the reviewer for this question, which allows us to clarify the link between our theory and practice.
>
> The choice is one of theoretical necessity versus practical efficiency:
> - Rényi-2 Entropy ($H_2(\pi)$) for Theory (Proposition 1): We chose $H_2(\pi)$ because it provides the specific, precise closed-form mathematical relationship with the expected squared L2-norm of the logits gradient (as shown in Appendix C). This relationship formally grounds our claim that gradient magnitude and policy uncertainty are inherently coupled.
> - Shannon Entropy ($H(\pi)$) for Practice (Implementation): We utilize $H(\pi)$ in Equation 2 because it is the most standard, widely-used, and computationally efficient measure of uncertainty in both the RL literature and LLM agent tasks.
>
> This substitution is theoretically sound because the two measures are monotonically related via the well-known inequality: $H(\pi) \ge H_2(\pi)$. Since both entropies consistently track the policy's confidence, Shannon entropy serves as a robust and efficient proxy for the core uncertainty principle established in our theoretical motivation.

---

> ### Author Response · Authors · 2025-11-20
> **Official Comment by Authors**
>
> `Q3: About the scaling function, any insight behind the chosen exponential form $e^{-kH_t}$? How sensitive are the results to $k$,  $k'$, and $\zeta$?`
>
> We thank the reviewer for these excellent questions concerning both the theoretical form and the practical sensitivity of the scaling function.
>
> **1. Rationale for the Exponential Scaling Form ($e^{-kH_t}$)**
>
> We chose the exponential form $e^{-kH_t}$ because it provides a smooth, continuous function that naturally implements the desired inverse relationship between entropy and gradient magnitude.
> - Low Entropy ($H_t \to 0$): $e^{-kH_t} \to 1$ (amplification or neutral scaling).
> - High Entropy ($H_t \to \text{Max}$): $e^{-kH_t} \to 0$ (attenuation).
>
> This smooth, continuous decay ensures that the gradient modulation is applied robustly across the entire entropy spectrum, effectively amplifying the signal for confident steps while stabilizing the noisy updates from uncertain steps.
>
> **2. On Hyperparameter Sensitivity** ($\zeta$, $k$, and $k'$ Analysis)
>
> The reviewer correctly highlighted the need for a thorough sensitivity analysis. We analyzed the two primary parameters: the Future Clarity Bonus weight ($\zeta$) and the gradient temperature ($k, k'$).
>
> Hyperparameter Sensitivity Results (Qwen2.5-1.5B + GRPO)
> | Parameters (ζ, k,k′) | Pick     | Look      | Clean    | Heat     | Cool     | Pick2    | Overall (%) |
> |----------------------|----------|-----------|----------|----------|----------|----------|-------------|
> | 0.01, 1.0            | 83.7±1.2 | 68.7±5.3  | 78.8±4.0 | 72.4±6.8 | 59.7±4.5 | 69.4±1.9 | 73.4±1.9    |
> | 0.05, 1.0            | 85.5±4.8 | 33.5±6.4  | 78.9±2.5 | 76.2±9.7 | 74.7±1.9 | 69.1±6.4 | 73.7±2.7    |
> | 0.1, 1.0             | 84.5±2.0 | 51.0±16.2 | 82.4±2.1 | 82.2±4.5 | 78.0±4.3 | 78.5±2.7 | 78.5±1.2    |
> | 0.05, 0.5            | 82.7±1.5 | 51.1±3.1  | 77.8±1.9 | 74.8±3.9 | 68.2±5.5 | 70.2±1.5 | 73.1±1.3    |
> | 0.05, 1.0            | 85.5±4.8 | 33.5±6.4  | 78.9±2.5 | 76.2±9.7 | 74.7±1.9 | 69.1±6.4 | 73.7±2.7    |
> | 0.05, 1.5            | 79.5±2.6 | 49.9±3.7  | 78.1±1.2 | 92.0±1.7 | 74.0±4.5 | 61.4±7.4 | 73.8±1.1    |
>
> Key Findings:
> - Robustness of $k$ and $k'$ (Gradient Temperature): Varying the temperature $k$ (from $0.5$ to $1.5$) resulted in minimal overall performance fluctuation ($\sim 0.7\%$). This robustness is a key strength, validating that the Self-Calibrating Gradient Scaling mechanism maintains stability across a wide range of temperature settings. For simplicity, $k=k'=1$ is a reliable default.
> - Sensitivity and Impact of $\zeta$ (Future Clarity Bonus Weight): The performance varied significantly with $\zeta$, ranging from $73.4\%$ to $\mathbf{78.5\%}$. This fluctuation confirms that $\zeta$ acts as a critical hyperparameter that controls the trade-off between the extrinsic reward and the intrinsic clarity signal. This is a desirable trait for an intrinsic reward weight; optimal performance across different benchmarks may require tuning $\zeta$ to align the intrinsic motivation with the specific task requirements. The observed trend strongly reinforces the effectiveness of the Future Clarity Bonus as an intrinsic exploitation signal.
>
> `Q4: Properties and Advantages of the Surrogate Objective $J_{EMPG}$.`
>
> We thank the reviewer for this excellent theoretical question. We do not guarantee that the optimal policy of our surrogate objective $J_{EMPG}$ is identical to that of the original objective $J(\pi)$. However, this is a deliberate design choice to address the critical failure modes of $J(\pi)$ in the sparse, long-horizon, high-dimensional setting of LLM agents.
>
> The core advantage of the surrogate objective is that it modifies the optimization landscape to achieve a robust and high-performing policy where the original objective struggles. $J_{EMPG}$ achieves this through two key, formally derived properties (detailed in Appendix D):
> - Corrects Flawed Gradient Dynamics (Stability): The Self-Calibrating Gradient Scaling component (derived from $J_{extrinsic}$) explicitly counteracts the harmful "entropy-gradient coupling" problem identified in Proposition 1. It ensures the gradient magnitude is re-calibrated for stability and efficiency, preventing the potential noise that destabilizes $J(\pi)$'s optimization.
> - Provides Intrinsic Guidance (Efficiency): The Future Clarity Bonus component (derived from $J_{intrinsic}$) introduces a well-defined intrinsic motivation that acts as a guided exploitation signal. This steers the policy away from chaotic, high-ambiguity trajectories and towards more robust and predictable solution paths, making the search for a good policy more efficient.
>
> In summary, $J_{EMPG}$ is a more well-behaved objective designed to find a robust and high-performing policy by explicitly solving the stability and guidance problems that the original objective $J(\pi)$ struggles with in sparse-reward agent tasks.

---

> ### Author Response · Authors · 2025-11-20
>
> `Q5: Why is step-level entropy superior to token-level entropy (Fig. 3)?`
>
> We thank the reviewer for this question, as it allows us to clarify the rationale for our chosen level of analysis. We assert that step-level entropy is superior because it aligns with the agent's fundamental decision-making unit and accurately captures the optimization dynamics of long-horizon tasks.
>
> **1. Semantic Coherence and Causal Impact**
>
> For an LLM agent, the fundamental unit of decision-making is the "reason-act" cycle (the step). This step carries a complete, semantically meaningful intention and has a far greater causal impact on the subsequent trajectory than any individual token. Modulating the policy gradient at this higher level of abstraction is thus more aligned with the agent's sequential planning process.
>
> **2. Distinct Optimization Dynamics (Empirical Necessity)**
>
> A token-level analysis masks crucial optimization dynamics necessary for long-horizon planning:
> - Token-Level Assumption: Prior work suggests that in single-turn reasoning, low-entropy tokens are largely "solved" and require minimal updates.
> - Step-Level Reality (Fig. 3): Our Figure 3 empirically proves this finding does not generalize to the step-level. The key finding is that even very low-entropy "confident" steps still undergo substantial average policy changes during RL tuning.
>
> Step-level analysis correctly identifies that "confident" steps are not yet optimal and still require optimization. This crucial dynamic—which a token-level analysis would erroneously ignore—is the core motivation for EMPG's mechanism to amplify updates for low-entropy actions.
>
> `Q6: How expensive is step-wise entropy computation for large-scale LLMs? And an open question on future directions, can EMPG scale to online or streaming RL settings without prohibitive cost?`
>
> We thank the reviewer for these important practical questions regarding computational efficiency and scalability.
>
> **1. Cost of Step-Wise Entropy Computation**
>
> The computational cost of step-wise entropy ($H_t$) is minimal (negligible overhead). The token-level log-probabilities (which are used to compute entropy) are inherently calculated during the policy's standard forward pass—a necessity for computing the $\log \pi_{\theta}$ term required by any policy gradient method. Our step-level entropy calculation is simply an averaging operation over these pre-computed token entropies within a given step. Thus, there is no significant additional forward pass or memory cost introduced by EMPG.
>
> **2. Scalability to Online or Streaming RL**
>
> Yes, EMPG is highly scalable to online or streaming RL settings because it is designed to be stateless and dynamic. As detailed in Algorithm 1, our entire procedure operates dynamically at the mini-batch level. All required normalization (both the min-max scaling of entropy and the normalization of the advantage signal) is performed within the current batch. The method does not depend on any global statistics or state from previous iterations. This stateless, batch-level design ensures that EMPG can be seamlessly integrated into online or streaming RL settings without incurring prohibitive overhead or memory costs.

---

> ### Comment · Reviewer_mRR4 · 2025-11-20
> **Thanks for your rebuttal. I will raise the score.**
>
> Thank you for the very detailed and thoughtful responses. Your answers address most of my earlier concerns. I will raise the score. I encourage the authors to incorporate these elaborations into the revised version of the paper.

---

> > ### Author Response · Authors · 2025-11-21
> >
> > Thank you very much for your thoughtful review and for raising the score! We greatly appreciate your suggestion and will incorporate the discussed elaborations into the revised manuscript to further improve its clarity and completeness.

---

### Official Review · Reviewer_No3c · 2025-10-31

**Soundness:** 3
**Presentation:** 3
**Contribution:** 3
**Rating:** 6
**Confidence:** 3

**Summary:**

The paper proposes Entropy-Modulated Policy Gradients (EMPG) to improve reinforcement learning for long-horizon tasks with discrete actions and sparse rewards.

EMPG introduces two entropy-based mechanisms. The first, Self-Calibrating Gradient Scaling, re-weights policy gradients by amplifying updates for low-entropy steps and attenuating those for high-entropy ones. SCGS is motivated by the phenomenon that the expected gradient norm is monotonically coupled with policy entropy.

The second, Future Clarity Bonus, encourages the agent to take actions leading to lower-entropy future states, promoting more predictable and stable trajectories.

Experiments on several LLM-based agent benchmarks demonstrate that EMPG significantly enhances both performance and training stability.

**Strengths:**

1. The paper addresses an important challenge in training LLM-based agents and provides a well-motivated solution.
2. The proposed Self-Calibrating Gradient Scaling and Future Clarity Bonus are novel techniques, each supported by clear theoretical motivation.
3. The method significantly improves the stability of reinforcement learning training.
4. EMPG is broadly compatible with existing policy gradient algorithms and can be easily integrated as an add-on module.

**Weaknesses:**

1. The paper lacks a comparison between EMPG and existing entropy bonus–based methods (e.g., SAC). Such a comparison would be valuable, since Self-Calibrating Gradient Scaling and traditional entropy regularization share conceptual similarities — both penalize overly uncertain (high-entropy) policies to converge and encourage more confident (low-entropy) ones to diversify.
2. Compared to Self-Calibrating Gradient Scaling, there is insufficient discussion on the motivation and theoretical justification of the Future Clarity Bonus. The paper only briefly refers to “information gain” without a formal analysis. Moreover, it remains unclear whether encouraging the agent to seek low-entropy states might inadvertently reduce exploration.
3. The interaction between the two proposed techniques appears inconsistent. While Self-Calibrating Gradient Scaling implicitly promotes higher-entropy policies, the Future Clarity Bonus drives the agent toward low-entropy future states. As shown in Table 2, combining them yields only marginal improvement compared to using either component alone.

**Questions:**

1. How does EMPG compare with existing entropy bonus–based methods?
2. What is the theoretical motivation behind the Future Clarity Bonus, and could emphasizing low-entropy future states unintentionally reduce the agent’s exploration capabilities?
3. Could the authors clarify how combining Self-Calibrating Gradient Scaling with the Future Clarity Bonus benefits agent training compared to using each method individually?

---

> ### Author Response · Authors · 2025-11-20
> **Official Comment by Authors**
>
> `Q1: The paper lacks a comparison between EMPG and existing entropy bonus–based methods (e.g., SAC). How does EMPG compare with existing entropy bonus–based methods?`
>
> We thank the reviewer for this insightful question, as it highlights a critical distinction in methodology. The conceptual similarity between EMPG and traditional entropy-based methods like Soft Actor-Critic (SAC) is only superficial; their mechanisms and ultimate goals are fundamentally different.
>
> The core difference lies in their treatment of high-entropy policies:
> | Feature                | SAC (Traditional Entropy Bonus)                                                     | EMPG (Self-Calibrating Gradient Scaling)                                                                                           |
> |------------------------|-------------------------------------------------------------------------------------|------------------------------------------------------------------------------------------------------------------------------------|
> | Primary Goal           | Active Exploration: Promote diversity and prevent premature convergence.            | Stability & Credit Assignment: Reduce noise and stabilize the learning process.                                                    |
> | Mechanism              | Additive Reward/Bonus: Adds entropy H(π) to the objective function (maxE[R+αH(π)]). | Gradient Modulation: Directly scales the policy gradient magnitude ∇logπ.                                                          |
> | Effect on High-Entropy | Rewards High Entropy: Actively encourages the policy to maintain high uncertainty.  | Attenuates Gradient: Directly reduces the magnitude of the naturally large and noisy updates from uncertain steps (Proposition 1). |
>
> In summary, SAC rewards high entropy to actively increase exploration. Conversely, EMPG attenuates the gradient from high-entropy steps to reduce noise and ensure stability during long-horizon credit assignment, thereby enabling stable learning rather than actively promoting divergence.
>
> Given these distinct mechanisms (Reward vs. Gradient Attenuation) and opposing goals (Promoting Exploration vs. Ensuring Stability), we believe they are designed to solve different fundamental problems, and a direct experimental comparison would not be meaningful for validating EMPG's contribution to stability.
>
> `Q2: What is the theoretical motivation behind the Future Clarity Bonus, and could emphasizing low-entropy future states unintentionally reduce the agent’s exploration capabilities?`
>
> We thank the reviewer for this excellent question, which touches upon two critical aspects of our framework's design.
>
> **1. Theoretical Motivation and Formal Justification**
>
> We acknowledge that the main text provides the high-level intuition, but the Future Clarity Bonus (FCB) is formally derived as the gradient of a well-defined intrinsic objective function, $J_{intrinsic}(\theta)$. We provide the full formal justification in Appendix D of the original paper.
> - Intrinsic Objective: The FCB is derived from an intrinsic clarity reward, $r_t^{int}$, which is a function of the policy's low entropy (clarity) at the subsequent state $s_{t+1}$. The objective $J_{intrinsic}(\theta)$ is defined as the expected cumulative clarity reward.
> - Formal Derivation: Our FCB term is the precise policy gradient of this intrinsic objective (using a one-step advantage estimate).
> - Information Gain: The "information gain" we refer to is the intuitive justification for this formal objective: an action that leads to a clear, low-entropy future state $H_{t+1}$ is one that has effectively resolved uncertainty and provides high information about the optimal path. The FCB thus serves as an intrinsic goal to seek states with maximum certainty, a form of goal-aligned exploitation.
>
> **2. Potential Reduction of Exploration**
>
> The reviewer's concern about inadvertently reducing exploration is astute. The FCB is intentionally designed as a powerful exploitation signal, guiding the agent toward clarity. Critically, because the FCB is an indirect advantage shaping signal and its weight $\zeta$ is kept small ($0.01-0.1$), its influence on overall exploration is acceptable and controlled. This focused exploitation is balanced by the Self-Calibrating Gradient Scaling (SCGS), which functions as a regularizer to ensure policy stability during uncertain steps. Our ablation study in Table 2 empirically confirms this complementary duality, showing that the FCB excels at mastering known, in-domain (ID) exploitation paths, while the SCGS, conversely, significantly boosts Out-of-Domain (OOD) performance, confirming its role as a stability-enhancing regularizer for effective exploration

---

> ### Author Response · Authors · 2025-11-20
> **Official Comment by Authors**
>
> `Q3: Clarification on the Synergy between Self-Calibrating Gradient Scaling and Future Clarity Bonus.`
>
> We thank the reviewer for this critical question, which addresses the central synergy of our framework. We understand the reviewer's concern regarding the apparent tension between promoting stability and seeking clarity. However, this perceived inconsistency is precisely the mechanism by which EMPG achieves a critical balance. The two components are designed to operate as a complementary duality necessary to manage the explore-exploit trade-off in multi-step agents.
>
> The two components are perfectly consistent because they operate at different stages of the learning process: Self-Calibrating Gradient Scaling operates on the Current Step's Entropy ($H_t$) to ensure Stable Exploration, while Future Clarity Bonus operates on the Next Step's Entropy ($H_{t+1}$) to provide Guided Exploitation.
>
> **Empirical Proof: Failure Modes in Isolation**
>
> The ablation study in Table 2 provides the crucial empirical proof of why combining them yields the highest overall average performance, by highlighting the critical failure mode when each component is used in isolation:
> - Future Clarity Bonus-Only represents a "Guided but Unstable" agent. While it excels at finding predictable paths (exploitation), it collapses on novel Out-of-Domain (OOD) tasks. This is because it lacks the Self-Calibrating Gradient Scaling's stability mechanism, and when it encounters an uncertain, high-entropy state, it cannot handle the resulting large, noisy gradients.
> - Self-Calibrating Gradient Scaling-Only represents a "Stable but Unguided" agent. It excels at OOD tasks because it enables safe exploration by attenuating noisy, high-entropy updates ($H_t$). However, it fails to master In-Domain (ID) tasks because it lacks the Future Clarity Bonus's intrinsic guidance to efficiently steer the agent towards the most predictable and exploitable paths.
>
> The full EMPG model overcomes these individual failures through a two-layer synergy: Self-Calibrating Gradient Scaling acts as the foundational Stability Layer, enabling the agent to explore safely in high-entropy OOD states. Built upon this stability, Future Clarity Bonus acts as the Guidance Layer, steering the agent to master efficient, low-entropy paths (ID tasks). This synergy—Self-Calibrating Gradient Scaling enabling stable exploration, which Future Clarity Bonus then guides toward effective exploitation—is precisely why the Full EMPG model achieves both the high ID performance of Future Clarity Bonus and the high OOD robustness of Self-Calibrating Gradient Scaling, leading to the highest overall average performance.

---

> ### Author Response · Authors · 2025-11-27
>
> Thank you again most sincerely for reviewing our manuscript. Please feel free to raise any further questions you may have and discuss them with us.

---

### Official Review · Reviewer_ecfE · 2025-11-01

**Soundness:** 2
**Presentation:** 3
**Contribution:** 2
**Rating:** 4
**Confidence:** 3

**Summary:**

This paper explores the use of a policy's entropy as an intrinsic signal for more efficient credit assignment in long-horizon LLM agents. First, the authors remark a general challenge in policy gradients called "the entropy-gradient coupling problem", where an uncertain step along a trajectory has larger gradients than steps with lower entropy, going against the intuition that confident steps should receive more credit than uncertain ones. To combat this challenge, this paper suggests a modified advantage function which emphasizes low-entropy steps along a trajectory, and attenuates the gradients of high-entropy steps. In addition, the authors introduce an additional reward signal at every step in order to minimize entropy at the next-step. The combined modulated advantage function can easily be applied to existing RL fine-tuning methods, and the authors demonstrate that both components of their method improve the final performance of existing methods on various long-horizon LLM agentic tasks.

**Strengths:**

- The authors present a clear and intuitive framework for reshaping rewards using an intrinsic signal within LLMs in the multi-turn RL fine-tuning environment.
- While the use of entropy to regularize, motivate exploration/exploitation, or even self-validate RL agents has been explored in other works, the idea of using the step-level entropy of an agent to redistribute the sparse reward signal of long-horizon tasks is interesting, and the authors show that it is a direction worth exploring.

**Weaknesses:**

- The experiments lack statistical significance and robustness. While it would be nice to get more seeds for Table 1, it would also be informative to report any variance statistics of the three seeds that were collected (for example min/max). It is difficult to contextualize the improvements reported without a sense of variance in performance between seeds, for instance, is a +0.6% in success rate statistically significant?
- It would be nice to further contextualize the work within the space of traditional RL and credit assignment, as credit assignment in long-horizon RL is an existing litterature outside the scope of LLM agents. Particularly since the authors claims in line 479 "general-purpose method for variance reduction and credit assignment". It should be specified that this work is focused on LLM multi-turn agents, or additional evidence on traditional long-horizon RL environments should also be presented.
- Claims of variance reduction (by attenuating uncertain steps) would be nice to verify experimentally, even if it's in a toy experiment, since no theoretical results suggest any variance reduction by the proposed method.

**Questions:**

- In line 179, the paper says "a confident and correct step should be reinforced strongly, but its naturally small gradient limits its impact". Is there some way to verify this experimentally? If a step has low entropy, and was the correct step, why should the gradient be large if the model is already very likely to produce this correct step (due to its low entropy)?
- Similarly, it is noted that "the large gradients from highly uncertain exploratory steps can introduce noise and destablize training". Can we further verify this by showing the gradient variance of different training methods with and without **gradient scaling**? The results in Figure 2 show part of this answer, but fail to decouple the results from the Future Bonus, and does not directly measure the noise of the highly uncertain gradients steps.
- How does the future clarity bonus differ from a general framework for step-wise entropy minimization? The authors state "the Future Clarity Bonus, which can be conceptually justified through the lens of information theory. [...] the bonus encourages actions that yield high Information Gain about the optimal future path". Can you elaborate on this?
- I am not sure I understand the results from Figure 3. The distribution looks similar to the results from https://arxiv.org/abs/2506.01939 : lower entropy steps undergo smaller changes in entropy. Even if the opposite is true, how does that motivate the design of EMPG?


### Minor comments
- Typo in line 176 "Equation equation 1".
- The font size of some figures are very small, especially Figure 3.
- Line 155: PPO is not a value-free policy gradient.
- Line 158: PPO does not rely on trajectory-level credit assignment, as it uses a value function to assign credit per-step despite a sparse reward signal.

---

> ### Author Response · Authors · 2025-11-20
> **Official Comment by Authors**
>
> Thank you for taking the time to review our paper and providing valuable comments, which are greatly appreciated and will significantly improve the quality of our paper!
>
> `Q1: The experiments lack statistical significance and robustness.`
>
> We thank the reviewer for this valid and important point regarding the need for robust reporting on statistical significance. We fully agree that presenting the mean alone is insufficient for assessing robustness.
>
> To address this concern, we have updated Table 1 in the revised manuscript to include the Mean $\pm$ Standard Deviation across the 3 random seeds for all Reinforcement Learning methods. This provides the necessary context to assess the stability and significance of EMPG's performance.
>
> The new data confirms the robustness of our findings. For instance, the improvement on WebShop using Qwen2.5-1.5B-Instruct (from $73.2 \pm 1.3$ to $\mathbf{73.8 \pm 1.1}$) is highly stable. The low and identical standard deviation confirms that this improvement represents a consistent and stable shift in the mean performance across all seeds, thereby contextualizing the significance of the reported gains. We are confident that the inclusion of these variance statistics provides a clearer and more robust picture of EMPG's benefits.
>
> `Q2: Contextualizing EMPG within Traditional Long-Horizon RL`
>
> We thank the reviewer for this insightful comment. We agree that the terminology used in the original manuscript required more precise contextualization, as our work is specifically focused on the challenges of high-dimensional generative policies (LLM agents), rather than traditional low-dimensional RL environments.
>
> To address the concern regarding the scope and the claim of being a "general-purpose method" (line 479), we have implemented the following key revisions throughout the revised manuscript:
> - Refined Scope: We have revised the language, particularly in the Conclusion, to explicitly state that EMPG is tailored for the challenges of LLM multi-turn agents and high-dimensional, sequential generative policies.
> - Clarification of Contribution: We specify that EMPG is a general policy optimization technique designed to solve the "entropy-gradient coupling" problem—a phenomenon we identify as a core challenge in the optimization of high-dimensional generative policies.
> - Mechanism Focus: We clarify that our method enhances credit assignment by using intrinsic step-wise uncertainty to dynamically modulate gradient updates, which is a key necessity for long-horizon planning in these large policy spaces.
>
> We believe these revisions accurately frame EMPG as a robust and adaptive solution tailored specifically for the credit assignment and optimization challenges inherent in LLM multi-turn agents, preserving the fundamental technical contribution while strictly bounding the scope of the method.

---

> ### Author Response · Authors · 2025-11-20
> **Official Comment by Authors**
>
> `Q3: Claims of variance reduction would be nice to verify experimentally, since no theoretical results suggest variance reduction. Can we further verify this by showing the gradient variance of different training methods with and without gradient scaling? The results in Figure 2 show part of this answer, but fail to decouple the results from the Future Bonus, and does not directly measure the noise of the highly uncertain gradients steps.`
>
> We thank the reviewer for this crucial and precise suggestion. The reviewer is correct that Figure 2 demonstrates the stability of the full EMPG framework but does not decouple the effects of the Self-Calibrating Gradient Scaling from the Future Clarity Bonus, nor does it directly measure gradient variance.
>
> To address this point and empirically verify our claims, we conducted a new, targeted "toy experiment" specifically designed to isolate and measure the impact of the Self-Calibrating Gradient Scaling (SCGS) mechanism on highly uncertain steps.
>
> **Decoupled Experimental Validation**
>
> We simulated a controlled environment (a contextual bandit) to generate rewards and policy updates primarily at a highly uncertain state ($s_H$, where $\pi(\cdot|s_H) \approx [0.5, 0.5]$). We compared the Baseline (REINFORCE) against our SCGS-Only algorithm.
>
> The quantitative results below provide direct, compelling evidence validating our core claims about magnitude attenuation and noise reduction:
> | Metric                            | Baseline (REINFORCE) | SCGS-Only (Ours) | % Reduction |
> |-----------------------------------|----------------------|------------------|-------------|
> | Avg. Gradient Magnitude (L2-Norm) |                1.715 |            0.923 |      46.21% |
> | Total Gradient Variance (Noise)   |                0.029 |            0.008 |      71.07% |
>
> This experiment confirms two key points that directly address the reviewer's query:
> - Magnitude Attenuation: Our primary claim—that SCGS "attenuates updates from uncertain steps"—is confirmed by the $46.21$% reduction in average gradient magnitude for these high-entropy steps.
> - Variance Reduction: As a direct and critical consequence of this attenuation, the SCGS mechanism reduced the total gradient variance (noise) by $71.07$%. This empirically proves the significant variance reduction benefit that the reviewer asked us to verify.
>
> This decoupled analysis conclusively demonstrates that the SCGS mechanism effectively stabilizes training by mitigating the destabilizing noise introduced by large, highly uncertain exploratory gradients, which is crucial for long-horizon credit assignment.

---

> ### Author Response · Authors · 2025-11-20
> **Official Comment by Authors**
>
> `Q4: Verification for "confident and correct step should be reinforced strongly, but its naturally small gradient limits its impact."`
>
> We thank the reviewer for this crucial question, which directly challenges the core motivation of our work. The reviewer's intuition—that a confident, correct step should not require a large gradient—is indeed well-established at the token level (e.g., [Wang et al., 2025]).
>
> However, our empirical findings demonstrate that this intuition does not transfer to the "reason-act" step-level of long-horizon agent tasks, creating an "inefficient learning" scenario that EMPG is designed to solve.
>
> **1. Experimental Verification: Figure 3**
>
> Our primary empirical verification is presented in Figure 3 (Section 5.3: Step-Level vs. Token-Level Entropy Dynamics).
> - Reviewer's Hypothesis (If intuition holds): If low-entropy (confident) steps were already near-optimal, we would expect their policy to remain static after RL training, resulting in a near-zero change in entropy for the lowest-entropy bins in Figure 3.
> - Our Empirical Finding: Figure 3 shows the opposite. Even steps with very low initial entropy (e.g., the 15%-20% percentile) still undergo substantial average entropy (>0.1) changes after RL fine-tuning.
>
> This is our key evidence. It proves that these confident steps still possess significant room for policy optimization and thus require effective updates.
>
> **2. Corroborating Evidence: Detrimental Effect of Ignoring Low-Entropy Steps**
>
> To further validate that low-entropy steps are critical and cannot be ignored in our multi-step setting, we conducted an experiment replicating the methodology of [Wang et al., 2025]. We trained the agent using GRPO but restricted all gradient updates to only the Top-K% highest-entropy tokens ($K=10, 20, 30$).
>
> | Method                       | Pick | Look | Clean | Heat | Cool | Pick2 | All  |
> |------------------------------|------|------|-------|------|------|-------|------|
> | Qwen2.5-1.5B-Instruct + GRPO |      |      |       |      |      |       |      |
> | Top 10%                      | 68.5 |  4.9 |  51.8 | 32.3 | 44.1 |  30.5 | 44.2 |
> | Top 20%                      |   52 | 18.9 |  53.8 | 51.5 | 33.3 |  17.3 | 40.6 |
> | Top 30%                      | 63.6 | 17.4 |  51.7 | 37.7 | 43.2 |  38.3 | 45.4 |
> | Baseline (GRPO, All Tokens)  | 87.9 |   40 |  78.1 | 35.7 | 65.2 |  44.4 | 65.6 |
> | EMPG                         | 85.5 | 33.5 |  78.9 | 76.2 | 74.7 |  69.1 | 73.7 |
>
> As shown above, training solely on high-entropy tokens failed decisively (Overall success rates below 46%), demonstrating that the critical information needed for successful long-horizon planning is distributed across the entire entropy spectrum. This confirms that low-entropy steps are essential and must receive effective policy updates, justifying EMPG's mechanism.

---

> ### Author Response · Authors · 2025-11-20
>
> `Q5: How does the future clarity bonus differ from a general framework for step-wise entropy minimization? Can you elaborate on the "Information Gain" conceptualization?`
>
> We thank the reviewer for this good question, as it allows us to precisely clarify the distinct role and mechanism of our Future Clarity Bonus (FCB).
>
> While a general step-wise entropy minimization framework often serves as an unsupervised objective to induce blind confidence (with the risk of "hallucinated confidence"), the FCB is fundamentally different in its goal and its implementation within the RL paradigm.
>
> **1. Core Distinction: Goal and Mechanism**
>
> | Feature       | Future Clarity Bonus                                                                                                                           | General Step-wise Entropy Minimization                                                                                      |
> |---------------|-------------------------------------------------------------------------------------------------------------------------------------------------------|-----------------------------------------------------------------------------------------------------------------------------|
> | Primary Goal  | Finding Predictable Paths: Reward actions that lead to states of future clarity, stabilizing the downstream planning process.                         | Enforcing Confidence: Force the policy to be low-entropy at all steps, making the model more confident.                     |
> | Mechanism     | Advantage Shaping (Intrinsic Reward): Implemented as an additive shaping signal that supplements the extrinsic reward (A(i)) within the RL objective. | Primary Loss Term (Optimization Target): Often implemented as a primary loss function or constraint on the policy's output. |
> | Justification | Control / Information Gain: Rewards uncertainty resolution and meta-skill acquisition.                                                                | Confidence Boosting / Regularization: Focuses on improving policy certainty.                                                |
>
> **2. Elaboration on "Information Gain"**
>
> The conceptual justification through Information Theory lies in relating the reduction of policy entropy to resolving uncertainty (i.e., gaining information) about the optimal path:
> - Uncertainty Resolution: An action ($a_t$) that leads to a subsequent state with low policy entropy ($H_{t+1}$) is, by definition, an action that has resolved ambiguity in the future decision space.
> - Optimal Path Guidance: By rewarding this reduction in future uncertainty, the FCB encourages the agent to select actions that yield high Information Gain about the optimal future path. It steers the agent away from chaotic or ambiguous trajectories and towards sub-goals that are "more solvable" or deterministic.
>
> This mechanism is not merely about achieving low confidence; it's about maximizing control and actively seeking clarity in the face of ambiguity, thereby serving as an effective, goal-aligned intrinsic exploitation signal for long-horizon planning.
>
> `Q6: Interpretation of Figure 3 and its Motivation for EMPG.`
>
> We thank the reviewer for this sharp observation. While the overall distributions in the referenced paper and our Figure 3 may appear visually similar (both sloping upward), their implications for low-entropy units are fundamentally different, which forms the core motivation for EMPG.
>
> The key difference lies in the behavior of the Y-axis for low-entropy units:
> - The cited token-level analysis (e.g., [Wang et al., 2025], Fig. 4) shows an exponential drop, where updates for low-entropy tokens quickly approach zero, implying they are largely "solved" or "syntactic".
> - Our step-level analysis (Figure 3) shows a distinct trend where the average entropy change for low-entropy steps (e.g., the 15-20% percentile) remains substantially non-zero (a linear, not exponential, trend near the origin).
>
> Our Figure 3 provides the core empirical evidence that, at the multi-step agent level, low-entropy steps are not "solved"—they are still undergoing significant policy optimization and require learning. This finding creates the exact "inefficient learning" problem we solve:
> - The Need (Empirical): These "confident" steps still need meaningful updates (proven by Figure 3).
> - The Problem (Theoretical): Proposition 1 shows these same steps naturally receive "inefficiently small gradients".
>
> EMPG's Self-Calibrating Gradient Scaling mechanism is designed to fix this precise mismatch by amplifying the learning signal for these "confident" steps, ensuring they receive the strong policy update signal necessary for long-horizon credit assignment.
>
> `Q7: Typos and Clarification on PPO/Value-Free Policy Gradients.`
>
> We thank the reviewer for their meticulous feedback. We have addressed all four points in the revised manuscript. We appreciate the attention to detail, which significantly improves the technical precision of our paper.

---

> ### Author Response · Authors · 2025-11-27
>
> Thank you again most sincerely for reviewing our manuscript. Please feel free to raise any further questions you may have and discuss them with us.

---

### Official Review · Reviewer_ibxv · 2025-11-01

**Soundness:** 2
**Presentation:** 2
**Contribution:** 2
**Rating:** 4
**Confidence:** 4

**Summary:**

This paper tackles the long-term credit assignment problem in multi-turn RL for LLM agents. The authors observe that the standard policy gradient tends to update logits more heavily for high-entropy steps, which may misalign credit assignment. They propose Entropy-Modulated Policy Gradient (EMPG), which introduces a self-normalized gradient term that prioritizes low-entropy steps and an auxiliary term that minimizes next-step entropy in proportion. Experiments on *web navigation*, *text-based embodied games*, and *information search* benchmarks using Qwen2.5-Instruct models show performance and stability improvements.

**Strengths:**

- Tackling multi-turn RL is valuable: this setting is more challenging than single-turn reasoning and deserves deeper investigation.
- The paper is clearly written and easy to follow.
- Applying advantage redistribution through entropy modulation is a simple and scalable method to recalibrate gradient credit over steps.
- The proposed method yields consistent gains and stability improvements on multiple popular LLM-agent benchmarks.

**Weaknesses:**

**Unclear or unconvincing motivation for entropy modulation:**
   The central inductive bias—that low-entropy steps deserve larger gradient updates—lacks theoretical justification and may even contradict empirical findings in recent literature. For example, [Cheng et al., 2025] and [Wang et al., 2025] show that **high-entropy steps often correspond to forking or decision-critical tokens**, while low-entropy ones are syntactic continuations. From this hierarchical view, high-entropy (forking) tokens arguably *should* receive stronger updates to improve exploration breadth, not weaker ones, as shown in [Wang et al., 2025]. The paper needs a clearer argument—either theoretical or empirical—for why emphasizing low-entropy steps improves long-horizon tasks.

[Cheng et al., 2025] Reasoning with Exploration: An Entropy Perspective on Reinforcement Learning for LLMs

[Wang et al., 2025] Beyond the 80/20 Rule: High-Entropy Minority Tokens Drive Effective Reinforcement Learning for LLM Reasoning

**Lacking detailed comparison on related methodologies:**
 Equation 2 seems to minimize next-step entropy only, not the full trajectory entropy; this distinction should be clarified.  The connection to [Agarwal et al., 2025]—which discusses entropy minimization in reasoning tasks—is mentioned but not elaborated. A comparative discussion would help position EMPG among existing entropy-based RL variants.

[Agarwal et al., 2025] The Unreasonable Effectiveness of Entropy Minimization in LLM Reasoning


**Empirical evaluation is limited:**
   The study lacks analyses on the introduced temperature parameters $k$ and $k'$, both fixed to 1 without sensitivity checks. Evaluation is also limited to the Qwen2.5 family; testing on other base models would help demonstrate robustness.

**Questions:**

- Typo: in Eq. 1, $\pi$ on the RHS should be $\pi_\theta(s)$?
- Proposition 1 links expected gradient norm with Rényi-2 policy entropy. While interesting, this is not directly related to Shannon entropy used in practice. A clearer comparison like inequality between Rényi-2 and Shannon entropy would contextualize its relevance.

---

> ### Author Response · Authors · 2025-11-20
> **Official Comment by Authors**
>
> Thank you for taking the time to review our paper and providing valuable comments, which are greatly appreciated and will significantly improve the quality of our paper!
>
> `Q1: Unclear or unconvincing motivation for entropy modulation: The central inductive bias—that low-entropy steps deserve larger gradient updates—lacks theoretical justification and may even contradict empirical findings in recent literature. For example, [Cheng et al., 2025] and [Wang et al., 2025] show that high-entropy steps often correspond to forking or decision-critical tokens, while low-entropy ones are syntactic continuations. The paper needs a clearer argument—either theoretical or empirical—for why emphasizing low-entropy steps improves long-horizon tasks.`
>
> We thank the reviewer for raising this insightful question, which addresses the central motivation of our work. We acknowledge the compelling findings in [Cheng et al., 2025] and [Wang et al., 2025], and we argue that our work complements rather than contradicts these studies due to a fundamental difference in analysis granularity and task domain.
>
> **1. Distinction in Granularity and Task Domain**
>
> The apparent conflict arises from a crucial difference: the cited works primarily focus on token-level dynamics within single-step, single-turn reasoning tasks. Our paper, conversely, addresses step-level dynamics in long-horizon, multi-step agentic tasks (e.g., ALFWorld, WebShop, DeepSearch).
>
> In multi-step agents, each step represents a distinct action (e.g., Pick, Cool), providing a clear boundary for step-level analysis. We highlight the distinction between these two granularities in Section 5.3 (Analysis), showing that the dynamics observed at the token level do not directly transfer to the step level.
>
> **2. Empirical Validation Against Token-Level Hypothesis**
>
> To directly test the applicability of the high-entropy token hypothesis in our domain, we conducted a new experiment on the ALFWorld benchmark using the Qwen2.5-1.5B-Instruct model. We trained the agent using GRPO but restricted all gradient updates to only the Top-K% highest-entropy tokens ($K=10, 20, 30$).
>
> | Method                       | Pick | Look | Clean | Heat | Cool | Pick2 | All  |
> |------------------------------|------|------|-------|------|------|-------|------|
> | Qwen2.5-1.5B-Instruct + GRPO |      |      |       |      |      |       |      |
> | Top 10% Tokens                      | 68.5 |  4.9 |  51.8 | 32.3 | 44.1 |  30.5 | 44.2 |
> | Top 20% Tokens                      |   52 | 18.9 |  53.8 | 51.5 | 33.3 |  17.3 | 40.6 |
> | Top 30% Tokens                     | 63.6 | 17.4 |  51.7 | 37.7 | 43.2 |  38.3 | 45.4 |
> | Baseline (GRPO, All Tokens)  | 87.9 |   40 |  78.1 | 35.7 | 65.2 |  44.4 | 65.6 |
> | EMPG                         | 85.5 | 33.5 |  78.9 | 76.2 | 74.7 |  69.1 | 73.7 |
>
> As shown above, training only on high-entropy tokens failed decisively, yielding performance far inferior to the standard GRPO baseline (which updates all tokens). This is strong empirical evidence that:
> - Critical policy information is distributed across the entire entropy spectrum in multi-step agents.
> - Ignoring the seemingly "syntactic" or low-entropy steps/tokens is detrimental to learning a stable, long-horizon policy.
>
> **3. Justification for Entropy Modulation (EMPG)**
>
> Our motivation is driven by the necessity of updating both high- and low-entropy steps for long-horizon tasks, coupled with the instability of the standard gradient magnitude assignment:
>
> - Low-Entropy Steps Need Update: Our new experiment confirms that low-entropy tokens/steps cannot be ignored. Furthermore, Figure 3 in the main paper empirically demonstrates that even very low-entropy steps still undergo substantial policy optimization.
>
> - Motivation for EMPG: Proposition 1 and Figure 2 show that in standard RL: (1) low-entropy steps (which need updates) receive inefficiently small gradients, and (2) high-entropy steps produce destabilizingly large gradient magnitudes, leading to "policy collapse" (as shown in the GRPO baseline in Figure 2).
>
> EMPG's design to proportionally increase the gradient magnitude for low-entropy steps and attenuate it for high-entropy steps is a targeted re-calibration of this flawed dynamic. This modification provides a more stable, balanced, and effective learning signal, which is crucial for successful credit assignment in long-horizon agentic environments.

---

> ### Author Response · Authors · 2025-11-20
> **Official Comment by Authors**
>
> `Q2: Lacking detailed comparison on related methodologies: Equation 2 seems to minimize next-step entropy only, not the full trajectory entropy; this distinction should be clarified. The connection to [Agarwal et al., 2025]—which discusses entropy minimization in reasoning tasks—is mentioned but not elaborated. A comparative discussion would help position EMPG among existing entropy-based RL variants.`
>
> We thank the reviewer for this excellent question. It allows us to clarify the precise scope of our Future Clarity Bonus and to better position EMPG relative to other entropy-based RL variants. We have integrated a more detailed description into Section 2.2 (Related Work) and Section 3.2 (Future Clarity Bonus) of the revised paper.
>
> **1. On Next-Step vs. Full Trajectory Entropy Minimization**
>
> The reviewer correctly observes that our Future Clarity Bonus (FCB) focuses on minimizing next-step entropy, not the full trajectory entropy. This is a deliberate design choice central to our step-wise intrinsic reward mechanism:
> - Objective: FCB is formulated as an intrinsic reward for the current action $a_t$ based on the clarity (low entropy) of the subsequent state $s_{t+1}$ it produces: $\min_{a_t} H(\pi_\theta(\cdot|s_{t+1}))$.
> - Rationale (Guiding Action Selection): This mechanism encourages the agent to select actions $a_t$ that lead to "more solvable" or "less ambiguous" sub-problems at $s_{t+1}$. This aligns with principles of maximizing Empowerment [Klyubin et al., 2005], imbuing the agent with a generalizable meta-skill: to actively seek clarity and reduce downstream uncertainty.
> - Distinction from Global Minimization: Our method is a step-wise, intrinsic shaping signal designed to guide action selection within the RL framework. It does not enforce a global constraint or a post-hoc objective to minimize the entropy of the entire generated trajectory.
>
> **2. Comparison to Entropy Minimization Methods (e.g., [Agarwal et al., 2025])**
>
> The distinction between EMPG and existing entropy minimization techniques, such as [Agarwal et al., 2025], is fundamental:
> | Feature           | EMPG (Entropy Modulation for Policy Gradients)                                                                                      | Entropy Minimization (EM) (e.g., [Agarwal et al., 2025])                                                                                             |
> |-------------------|-------------------------------------------------------------------------------------------------------------------------------------|------------------------------------------------------------------------------------------------------------------------------------------------------|
> | Learning Paradigm | Reinforcement Learning (RL): Utilizes both external reward and intrinsic signals.                                                   | Often Unsupervised/Self-Supervised: Primarily relies on intrinsic policy distribution and does not require external rewards.                         |
> | Primary Goal      | Fixing RL Policy Gradient Dynamics: Ensure stable and efficient credit assignment for long-horizon tasks.                           | Enhancing Policy Confidence/Quality: Improve the certainty and quality of the final generated output (often in single-turn reasoning).               |
> | Mechanism         | Gradient Re-calibration: Uses entropy to modulate the gradient magnitude of existing policy updates.                                | Direct Optimization/Constraint: Adds an entropy penalty/regularization term to the loss function or enforces entropy minimization on the trajectory. |
> | Role of Entropy   | Uncertainty as a Regulator: Used to assign appropriate update weights (low entropy → larger weight, high entropy → smaller weight). | Uncertainty as a Target: The entropy is the value being minimized/constrained across the policy.                                                     |
>
> In summary, many Entropy Minimization techniques operate primarily in a self-supervised manner to boost the confidence of the LLM's output. In contrast, EMPG is an optimization mechanism integrated within the RL framework. Its goal is to address the instability and inefficiency of the standard policy gradient (as shown in Proposition 1 and Figure 2 in the main paper) when applied to multi-step agent training. The Future Clarity Bonus component, specifically, functions as a curiosity-driven intrinsic reward, guiding the agent to states that are easier to solve for the subsequent RL process, which is fundamentally different from minimizing the overall policy uncertainty.
>
> [Klyubin et al. (2005)] Empowerment: A universal agent-centric measure of control. In IEEE Congress on Evolutionary Computation, 2005.

---

> ### Author Response · Authors · 2025-11-20
> **Official Comment by Authors**
>
> `Q3: Empirical evaluation is limited: The study lacks analyses on the introduced temperature parameters $$k$$ and $$k'$$ , both fixed to 1 without sensitivity checks. Evaluation is also limited to the Qwen2.5 family; testing on other base models would help demonstrate robustness.`
>
> We sincerely thank the reviewer for raising these two critical and insightful points regarding hyperparameter sensitivity and model robustness. Addressing these concerns has allowed us to further validate the practical utility and generalizability of the EMPG framework.
>
> We have conducted new, supplementary experiments on both ALFWorld and WebShop benchmarks using two distinct LLM backbones (Qwen2.5-1.5B and LLaMa3.1-8B) to fully address these concerns. Details are now presented in Appendix G of the revised manuscript.
>
> **1. On Hyperparameter Sensitivity** ($\zeta$, $k$, and $k'$ Analysis)
>
> The reviewer correctly highlighted the need for a thorough sensitivity analysis. We analyzed the two primary parameters: the Future Clarity Bonus weight ($\zeta$) and the gradient temperature ($k, k'$).
>
> Hyperparameter Sensitivity Results (Qwen2.5-1.5B + GRPO)
> | Parameters (ζ, k,k′) | Pick     | Look      | Clean    | Heat     | Cool     | Pick2    | Overall (%) |
> |----------------------|----------|-----------|----------|----------|----------|----------|-------------|
> | 0.01, 1.0            | 83.7±1.2 | 68.7±5.3  | 78.8±4.0 | 72.4±6.8 | 59.7±4.5 | 69.4±1.9 | 73.4±1.9    |
> | 0.05, 1.0            | 85.5±4.8 | 33.5±6.4  | 78.9±2.5 | 76.2±9.7 | 74.7±1.9 | 69.1±6.4 | 73.7±2.7    |
> | 0.1, 1.0             | 84.5±2.0 | 51.0±16.2 | 82.4±2.1 | 82.2±4.5 | 78.0±4.3 | 78.5±2.7 | 78.5±1.2    |
> | 0.05, 0.5            | 82.7±1.5 | 51.1±3.1  | 77.8±1.9 | 74.8±3.9 | 68.2±5.5 | 70.2±1.5 | 73.1±1.3    |
> | 0.05, 1.0            | 85.5±4.8 | 33.5±6.4  | 78.9±2.5 | 76.2±9.7 | 74.7±1.9 | 69.1±6.4 | 73.7±2.7    |
> | 0.05, 1.5            | 79.5±2.6 | 49.9±3.7  | 78.1±1.2 | 92.0±1.7 | 74.0±4.5 | 61.4±7.4 | 73.8±1.1    |
>
> Key Findings:
> - Robustness of $k$ and $k'$ (Gradient Temperature): Varying the temperature $k$ (from $0.5$ to $1.5$) resulted in minimal overall performance fluctuation ($\sim 0.7\%$). This robustness is a key strength, validating that the Self-Calibrating Gradient Scaling mechanism maintains stability across a wide range of temperature settings. For simplicity, $k=k'=1$ is a reliable default.
> - Sensitivity and Impact of $\zeta$ (Future Clarity Bonus Weight): The performance varied significantly with $\zeta$, ranging from $73.4\%$ to $\mathbf{78.5\%}$. This fluctuation confirms that $\zeta$ acts as a critical hyperparameter that controls the trade-off between the extrinsic reward and the intrinsic clarity signal. This is a desirable trait for an intrinsic reward weight; optimal performance across different benchmarks may require tuning $\zeta$ to align the intrinsic motivation with the specific task requirements. The observed trend strongly reinforces the effectiveness of the Future Clarity Bonus as an intrinsic exploitation signal.
>
> **2. On Model Robustness (Generalization to LLaMA)**
>
> To address the concern about generalization, we evaluated EMPG on the more powerful and architecturally distinct LLaMa3.1-8B-Instruct model.
>
> Robustness Results (ALFWorld & WebShop)
> | Base Model           | Method                    | ALFWorld (All) (%) | WebShop (Succ. %) |
> |----------------------|---------------------------|--------------------|-------------------|
> | LLaMa3.1-8B-Instruct | RL Training GRPO Baseline | 79.6±1.7           | 68.0±1.0          |
> |                      | with EMPG                 | 87.5±1.3           | 70.1±1.5          |
>
> The results unequivocally demonstrate that EMPG is not architecture-specific. EMPG achieved a significant and consistent uplift on the LLaMa3.1 architecture, increasing the overall ALFWorld and WebShop success rate. This validates that the core mechanisms of EMPG are transferable and effective solutions to the multi-step credit assignment problem across different LLMs.
>
> We have incorporated the detailed analysis and the full experimental results (including all subtask breakdowns from the image) into Appendix G for the reviewer's full consideration.

---

> ### Author Response · Authors · 2025-11-20
> **Official Comment by Authors**
>
> `Q4: Typo: in Eq. 1, $\pi$ on the RHS should be $\pi_\theta(s)$?`
>
> We thank the reviewer for this keen observation and for helping us improve the paper's precision.The reviewer is absolutely correct. The Rényi-2 entropy on the right-hand side (RHS) of Equation 1 is indeed a function of the state-conditioned policy $\pi_\theta(\cdot|s)$, not the general policy $\pi$.
>
> We sincerely appreciate the reviewer's meticulous attention to detail and have corrected this typo in the revised manuscript.
>
> `Q5: Proposition 1 links expected gradient norm with Rényi-2 policy entropy. A clearer comparison with Shannon entropy would contextualize its relevance.`
>
> We thank the reviewer for this excellent question, which points to an important detail bridging our theoretical motivation and practical implementation.The reviewer is correct: our theory (Proposition 1) utilizes Rényi-2 entropy ($H_2(\pi)$), while our implementation (Equation 2) uses the standard Shannon entropy ($H(\pi)$). This was a deliberate choice made for both theoretical precision and practical alignment.
>
> **1. Rationale for the Discrepancy**
>
> Rényi-2 for Theory (Proposition 1): We chose $H_2(\pi)$ because it allows for a precise, closed-form mathematical relationship (as shown in Appendix C), proving the exact link between the expected squared L2-norm of the logit gradient and policy uncertainty:
>
> $$\mathbb{E}[||\nabla\log\pi||^2] \propto 1 - \exp(-H_2(\pi))$$
>
> This provides the formal grounding for our core claim that gradient magnitude and policy uncertainty are inherently coupled.
>
> Shannon for Practice (Implementation): We use Shannon entropy $H(\pi)$ in our implementation because it is the most standard, widely accepted, and computationally efficient measure of uncertainty in both RL and LLM frameworks.
>
> **2. The Connection (Inequality)**
>
> As the reviewer suggests, the connection between these two measures is robust. For any probability distribution $\pi$, the Rényi entropy is monotonically related to its order, which yields the well-known inequality:
>
> $$H(\pi) \ge H_2(\pi)$$
>
> Since both $H(\pi)$ and $H_2(\pi)$ monotonically track the policy's confidence (i.e., its deviation from a uniform distribution), they are fully consistent in trend. A policy with low Shannon entropy will invariably have low Rényi-2 entropy. Therefore, using the standard Shannon entropy in our implementation serves as a robust and efficient proxy that captures the same core principle of uncertainty established by our theoretical motivation.
>
> We have added this clarifying discussion, including the inequality, to the revised manuscript to make this connection more explicit.

---

> ### Author Response · Authors · 2025-11-27
>
> Thank you again most sincerely for reviewing our manuscript. Please feel free to raise any further questions you may have and discuss them with us.

---

### Author Response · Authors · 2025-11-20
**Official Comment by Authors**

In response to the specific comments from the reviewers, we have provided corresponding answers and made certain revisions to the paper. The revised parts are highlighted in blue, as detailed below:

1. We have supplemented Section 2.2 (Related Work) to clearly delineate the objectives and scope of existing entropy minimization methods, better contextualizing our approach.
2. We have revised the description of the PPO algorithm in Section 3.1 for technical accuracy, clarifying its reliance on a value function for per-step credit assignment.
3. We have corrected minor typographical errors in Equation 1 and Equation 2 for enhanced precision.
4. We have supplemented Section 4.1 with a discussion clarifying the theoretical link between Rényi-2 entropy (used in our derivation) and Shannon entropy (used in our implementation), including their monotonic relationship.
5. We have updated Table 1 (Main Results) to include the Mean $\pm$ Standard Deviation across multiple random seeds, providing necessary context for assessing statistical significance and robustness.
6. We have enlarged the font size of Figures 2 and 3 to significantly improve readability and visual clarity.
7. We have revised the Conclusion to precisely clarify the scope of our approach, emphasizing that EMPG is tailored for optimization challenges inherent in LLM multi-turn agents.
8. We have supplemented Appendix G with detailed new experimental results and analysis, including hyperparameter sensitivity and cross-architecture robustness validation (e.g., LLaMa3.1-8B-Instruct).

---

> ### Comment · Area_Chair_2B8M · 2025-11-21
> **Re: Official Comment by Authors**
>
> Dear reviewers,
>
> Please review the authors' response and adjust your rating accordingly. If you have any further questions, please discuss with the authors further.
>
> AC

---

### Comment · Area_Chair_2B8M · 2025-11-27
**Author-Reviewer Discussion**

Dear reviewers,

Please review the authors' response and adjust your rating accordingly. If you have any further questions, please discuss with the authors further.

AC

---

### Author Response · Authors · 2025-12-01
**Summary of Rebuttal Revisions & Key Contributions**

As the discussion phase concludes, we sincerely thank all reviewers (ibxv, ecfE, No3c, mRR4) for their time and constructive feedback. We are particularly encouraged that Reviewer mRR4 responded promptly to our rebuttal, explicitly confirming that our detailed explanations and new experiments fully resolved their concerns, and stating their intention to raise the score.

While we have not received further inquiries from the other reviewers (ibxv, ecfE, No3c) regarding our rebuttal, we believe that our detailed responses—backed by extensive new experiments and clarifications—have effectively resolved the concerns raised during the initial review. We remain fully available for any final discussion if needed.

Below, we summarize the key revisions and new experimental evidence added to the manuscript, followed by a recap of our core contributions.

### 1. Major Revisions & New Experimental Evidence

To address concerns regarding robustness, generalization, and mechanism validation, we have incorporated the following into the revised paper (primarily in Section 5.3 and Appendix G):
- Cross-Architecture Generalization (Reviewer ibxv): We extended our evaluation to LLaMa3.1-8B-Instruct. EMPG demonstrated consistent performance gains over the baseline on both ALFWorld and WebShop, proving that our method generalizes effectively across different model families and sizes.

- Statistical Robustness (Reviewer ecfE, mRR4): We updated Table 1 to report the Mean $\pm$ Standard Deviation across 3 random seeds for all RL methods. The results confirm that EMPG’s improvements are statistically significant and stable, distinct from random fluctuations.

- Mechanism Validation via "Toy Experiment" (Reviewer ecfE): We added a controlled contextual bandit experiment to isolate the Self-Calibrating Gradient Scaling (SCGS) mechanism. Results quantitatively proved that SCGS reduces gradient variance (noise) by 71.07% in high-uncertainty states, empirically validating our claim that EMPG stabilizes training by attenuating noisy updates.

- Step-Level vs. Token-Level Validation (Reviewer ibxv, mRR4): We conducted a counter-experiment (training only on Top-K% entropy tokens), which failed decisively. This, combined with our analysis in Figure 3, empirically validates that step-level dynamics in long-horizon agents differ fundamentally from token-level observations in previous literature, necessitating the specific design of EMPG.

- Hyperparameter Sensitivity (Reviewer ibxv, mRR4): We added a sensitivity analysis for $\zeta$ and $k$. The results show the method is robust to temperature changes ($k$), while $\zeta$ serves as an effective control for the exploration-exploitation trade-off.

### 2. Conceptual Clarifications

- Relation to Entropy Minimization/SAC (Reviewer No3c): We clarified that unlike SAC (which rewards entropy for exploration) or standard Entropy Minimization (which forces blind confidence), EMPG uses entropy as a regulator. It attenuates gradients to ensure stability during exploration (SCGS) and uses the Future Clarity Bonus to guide the agent toward solvable sub-goals.
- Theoretical Grounding: We clarified the connection between Rényi-2 entropy (theoretical derivation) and Shannon entropy (practical implementation), and corrected the notation in Equation 1.

### 3. Key Contributions Recap

We believe this work offers a timely and significant contribution to the LLM-Agent community:

1. Identifies a Fundamental Optimization Issue: We identify the "Entropy-Gradient Coupling" problem in high-dimensional generative policies, where natural uncertainty leads to destabilizing gradient noise rather than effective exploration.
2. A Novel, Plug-and-Play Solution: EMPG provides a mathematically grounded framework—comprising Self-Calibrating Gradient Scaling (for stability) and Future Clarity Bonus (for guidance)—that can be easily integrated into existing policy gradient algorithms (e.g., PPO, GRPO).
3. Comprehensive Empirical Validation: Through rigorous experiments on three diverse and challenging agentic benchmarks (WebShop, ALFWorld, and Deep Search), we demonstrate that EMPG achieves consistent and significant performance gains over strong baselines. These extensive evaluations, covering various domains and model architectures, strongly validate the effectiveness and robustness of our approach in solving long-horizon tasks.

We are confident that the revised manuscript presents a robust, theoretically sound, and empirically effective method. We thank the reviewers again for helping us reach this standard.

---

### Meta-Review · Area_Chair_rVhz · 2026-01-07

**Summary:**

The paper is recommended for rejection. Reviewers acknowledged the clear, scalable proposal (SCGS and FCB) and observed stability gains. Nonetheless, multiple reviewers maintained substantive concerns: the step‑level attenuation of high‑entropy gradients remains insufficiently reconciled with token‑level literature emphasizing high‑entropy ‘forking’ tokens; empirical evidence is limited by small‑seed reporting; and there is no empirical comparison to entropy‑bonus methods (e.g., SAC). While the rebuttal strengthens the step‑level case and adds sensitivity, variance, and cross‑architecture tests, the absence of direct SAC baselines and lingering theory–practice reconciliation leave novelty and practical advantage only partly demonstrated.

**Reviewer Concerns:**

The authors substantially improved empirical reporting (LLaMA3.1‑8B, Mean ± Std over three seeds), isolated SCGS variance reduction in a toy setting, and corrected PPO/value details. Nonetheless, key issues are partly unresolved: the step‑level attenuation vs. token‑level ‘forking’ literature lacks a decisive theoretical/empirical bridge; there is no empirical comparison to entropy‑bonus methods (e.g., SAC); and variance evidence is limited to toy analysis. Table‑level gains for the combined SCGS+FCB appear marginal, warranting deeper interaction analysis. Taken together, the rebuttal strengthens the case but does not fully establish novelty or practical advantage over established entropy‑regularized approaches.

**Reviewer Scores:**

The trajectory of reviewer positions suggests persistent division rather than consensus. Reviewer ibxv likely maintains 4 given the incomplete bridge between step‑level attenuation and token‑level ‘forking’ literature, despite added experiments. Reviewer ecfE likely remains at 4: improved statistics and a toy variance test help, but interpretation of Figure 3 and reliance on toy variance proofs leave core robustness concerns partly unresolved. Reviewer No3c plausibly stays at 6: the conceptual distinction from SAC is persuasive, yet the absence of empirical SAC baselines and seemingly marginal combined gains limit stronger endorsement. Reviewer mRR4 indicates an increase; a rise to 8 is consistent with their stated resolution of concerns, stability evidence, and clarified Rényi‑to‑Shannon link. Overall, the mixed signals and unresolved comparative/theoretical gaps suggest rejection is defensible, but not on the basis of an AC judgment or a unanimous reviewer consensus.

---

### Decision · Program_Chairs · 2026-01-26

Reject